# Bottom fishery impact generates tracer peaks easily confused with bioturbation traces in marine sediments

Stefan Forster[1], Claudia Runkel[1], Josephin Lemke[1], Laura Pülm[1], Martin Powilleit[1]

[1]University of Rostock, Institute for Biosciences - Marine Biology, D-18059 Rostock, Germany

*Correspondence to*: Stefan Forster (stefan.forster@uni-rostock.de)

**Abstract.** In the process of reworking sediments and thus shaping biogeochemical processes, marine bottom dwelling animals are thought to play a pivotal role in many benthic environments. Bioturbation (particle reworking) includes downward transport of particles into the sediment as a major process and is sometimes detected as sub-surface maxima (peaks) of specific particulate substances (tracers). Here we document that sub-surface peaks, such as those typically attributed to biological particle transport in sediments, may equally be generated by otter boards in bottom trawling fishery. Boards can generate tracer peaks whereby they scoop sediment from the surface, flip it over and deposit it onto the adjacent sea floor. These peaks are indistinguishable from those generated by benthic fauna burying surface material at sediment depth. We demonstrate this for the particle tracer chlorophyll a in silty sand from the Western Baltic Sea with fauna that generally does not burrow deep in a global comparison. Our inability to distinguish the driving processes generating the peaks indicates limits to our understanding of magnitude and spatial extend of bioturbation traces in this environment. It also poses a problem to the assessment of fishery resource use and benthic processes. However, based on natural fauna abundance, behavioural information and fishery intensity data, we identify macrofauna and not otter boards as the dominant cause for peaks at the sites investigated here.

## 1 Introduction

Bottom trawling introduces anthropogenic disturbance to the seafloor. Research addressing different aspects of this activity is accumulating for it causes partial destruction of benthic habitats and its biota (Sparks-McConkey and Watling, 2001; Watling and Norse, 1998), alters sediment structure both physically and in its granulometry (Oberle et al., 2016a; Bradshaw et al., 2012, 2021), suspends finer grain sizes from the bulk sediment and may affect contaminant deposits (Oberle et al., 2016b). Trawling affects ecosystem functions such as carbon storage (Epstein et al., 2022) and sediment integrity (de Juan et al., 2015) and additionally interacts with other pressures on the benthic ecosystem such as contaminant deposits or hypoxia (Oberle et al., 2016b; Bunke et al., 2019; van Denderen et al., 2022).

Investigations aiming to detect and quantify the effects of fishing gear at the seafloor face the difficulty that patterns may also stem from disturbances, natural or anthropogenic, other than trawling (Bunke et al., 2019). Deep reaching storms, particle reworking by bioturbating fauna, construction and dredging activity leave traces of disturbance at the seafloor as well.

Localization of the impact on the sea floor and sampling also poses a major problem. Both usually take place with limited spatial precision, which is why a majority of studies rely on statistically capturing average effects in areas of certain trawling intensities. Measures such as swept area ratio (SAR) of fishing intensity remain inaccurate in that they average bottom trawls over long periods (per year or per quarter) and relate the impact to comparatively large areas (several km²). However, the damage to the surface sediment by otter boards may be local. Studies investigating the change of vertical distribution of sediment constituents demonstrate that an effect of trawling can be the removal of surface sediment and considerable alterations of matter concentrations and processes in this sediment surface layer (Mestdagh et al., 2018; van de Velde et al., 2018; Morys et al., 2021). In an experimental dredge trawl Morys et al. (2021) found that sediment excavated to 2.5-3 cm depth piled up irregularly on the sides of the track.

Particle reworking by bioturbating organisms is an important aspect of transport for substances at and just below the sediment-water interface. While specific motions associated to their way of life (burrow construction, feeding, defecation) move particles in all spatial directions, at vastly different time intervals and over very different distances at any one time, the macroscopic pattern of the sum of these individual reworking events is mostly dealt with in simplified ways. A common differentiation describes numerous and small ("local") transport steps as an erratic, non-directional mixing process (analogous to diffusion) and observes "non-local" transports when directional transport over longer distances takes place and creates peaks in vertical concentration profiles of particles in the sediment (e.g. chlorophyll used as a particle tracer).

The interpretation of peaks in stable tracer distributions (glass beads, luminophores) or decaying tracer distributions (radioisotopes, chlorophyll) as signs of non-local transport is widespread (Wheatcroft et al., 1994; Blair et al., 1996; Meysman et al., 2003; Morys et al., 2016, 2017; Oberle et al., 2016b). Peaks are observed regardless of the persistence of the tracers used, in stable tracers or tracers that decay with time. Natural decay of the tracer chlorophyll allows to "look back in time" for 100 - 150 days when its peak concentration has declined to 25 % of its original value. Decay thus determines whether a peak will remain visible or if the event merges into the overall mixing which is usually dealt with as diffusion analogue. Chlorophyll as a particle tracer can therefore show relatively recent events of particle mixing only. Compiling data on the frequency of use and geographic coverage of studies employing different tracers, Solan et al. (2019) showed that next to radioisotopes the naturally occurring chlorophyll a molecule (Chl-a) is commonly used.

Transport of particles in sediments potentially always implies changes in availability, concentration or distribution of food (organic particles), contaminants and oxidising agents such as iron oxides, with potential effects on carbon burial and inorganic nutrient release, including potential feed-back of these rate changes on bioturbating macrofauna (Epstein et al., 2022; de Borger et al., 2020; van Denderen et al., 2022) It additionally affects dissolved electron acceptor distribution as concomitant fluid movement is inevitable. This fosters the interest in bioturbation as an important regulator for bacterial activity and diagenesis (Aller, 2014).

In the framework of research on impacts of trawling in the Fehmarn Belt area (FB), Western Baltic Sea (Fig. 1) (https://www.io-warnemuende.de/dam-mgf-baltic-sea-home.html) we measured the depth distribution of Chl-a in order to study bioturbation. The Western Baltic Sea also harbours seafloors among the most intensively trawled areas of the world (Amoroso et al., 2018). Trawling tracks in this area have been thoroughly analyzed (Schönke et al., 2022) and their biogeochemical signals interpreted by Rooze et al. (2024). Chlorophyll peaks detected here are usually attributed to *Arctica islandica*, the dominant reworking bivalve, or other biota of the community.

The ocean quahog, *Arctica islandica*, lives just below the sediment surface, from where it maintains contact with seawater via its short siphon (Winter, 1969). *A. islandica* is classified as a surface biodiffusor (also: surficial modifier/biodiffusor; Queirós et al., 2013) based on its surface dwelling activity. Its activity causes constant and random local transport of particles over short distances in the uppermost centimetres of the sediment (Kristensen et al., 2012; Queirós et al., 2013). The species also shows a behaviour known as "survival by metabolic suppression" induced by hypoxia, when it burrows to deeper horizons. Therefore, the ocean quahog is also considered a "downward conveyor" that translocates particles to depth by non-local transport (Kristensen et al., 2012; Morys et al., 2017). In Mecklenburg Bay, *A. islandica* accounts for up to 99% of the biomass representing the most important species below the halocline (Morys et al., 2017).

However, intense trawling raises the question of an alternative origin of the peaks. Previous studies (Oberle et al., 2016b), UW-video evidence, and a possible analogy to terrestrial soil turnover during ploughing triggered the idea of surface sediment subduction by otter boards. To our knowledge there is no evidence of this transport process in the literature.

Anticipating sediment displacement similar to Morys et al. (2021) we conducted an *ex situ* experiment to mimic otter board effects at the sediment- water interface. We subsequently performed an *in situ* experiment with trawling and immediate targeted sampling by scuba divers. Our aims were to i) mimic the genesis of altered Chl-a distributions experimentally, ii) compare these with peaks found in the field and potentially originating from commercial trawling and finally iii) discuss the likelihood and consequences of confusion of these peaks with those generated by bioturbation.

## 2 Material and Methods

We performed experiments *ex situ* in a mesocosm and *in situ* by setting trawl marks to provide proof of principle of mechanisms operating. We investigated the changes of particle distribution brought about by the mechanic impact of otter trawling.

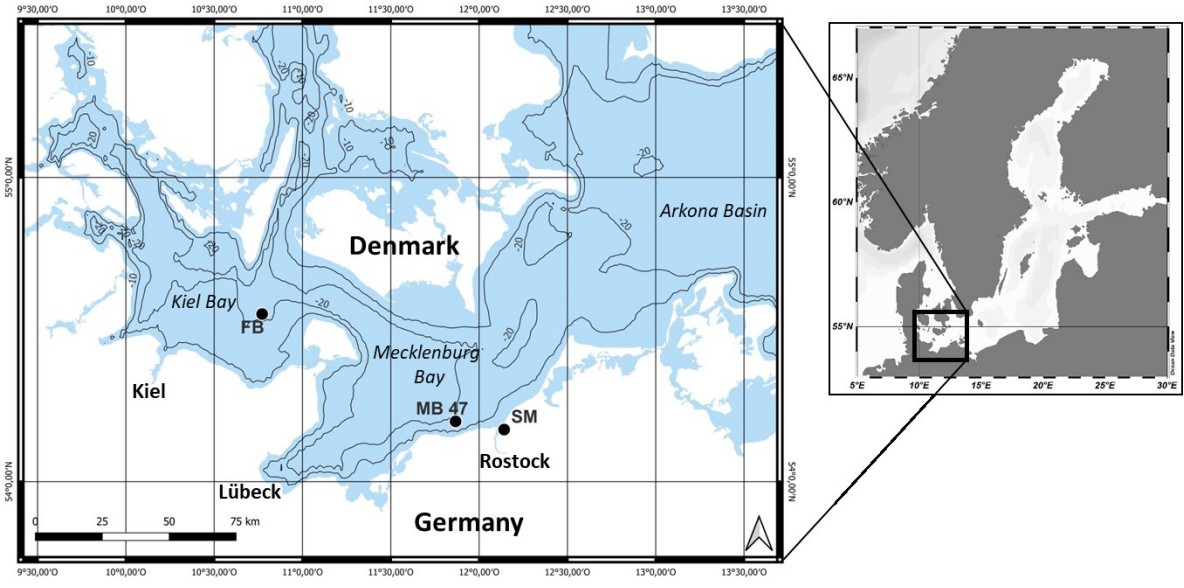

90

**Figure 1: Locations in the Western Baltic Sea where (1) random sampling revealed sub-surface tracer peaks in the field (FB), (2) an** *in situ* **trawling experiment was performed (MB 47) and (3) sediment was retrieved for an** *ex situ* **mesocosm experiment (SM).**

### *Ex situ* experiment

95    In April 2021 we simulated the mechanical trawling impact at a small scale in a mesocosm using a shovel and a rake. The mesocosm consisted of a circular aquarium (0.8 m diameter, 1 m height) which had been filled 3 month before with sandy sediment to reconstruct a horizontally homogeneous, vertically declining Chl-a distribution. The lowest sediment layer consisted of 15 cm sieved sand (0.5 mm) that had been stored in the dark for > 3 months. This was overlain by 8 cm freshly sieved sediment from the field site Schnatermann (SM, Fig. 1) that had been removed from 2 – 10 cm depth, excluding the

100   upper 2 cm surface layer sediment with much microphytobenthos at this shallow location (SM, 0.5 m water depth). Finally, the uppermost layer in the mesocosm contained 2 cm of sieved surface material from that field site harboring a rich microphytobenthos community. The upper 10 cm of sediment consisted of silty fine sand (m.d. 190 µm) similar to the sediment at the *in situ* experimental site. The mesocosm stood outside the university buildings at ambient light and temperature (temperature range 2 to 16 °C) and was covered by 15 cm of water (10 psu).

105   On April 8 2021, we manipulated the surface sediment with a shovel (~8 cm wide) and a rake according to the scheme in figure 2. We decided to excavate sediment keeping a defined geometry of depth and width when removing sediment rather than some plough-like tool which we found difficult to implement. In fact, we are not sure about the exact mechanism and geometry of sediment removal and its deposition on the adjacent sediment by an otter board, therefore the method used in the mesocosm is

a surrogate and not necessarily the same as the physical process that might be active *in situ*. We excavated sediment with a flat rectangular shovel, scoop by scoop to about 4 cm depth from the surface (track leading along sediment core position 6 – 10) and deposited it upside down to the left onto the adjacent sediment surface (from position 11 – 15). The resulting "furrow" and "mound" of about 40 cm length showed a surface topography with wave-like cross section as shown in figure 2 (dashed line). We also pulled a rake through the sediment along the core positions 1 – 5 mixing the sediment to ~ 2 cm depth while slowly moving up and down. This is to mimic the impact of the trawl net with its footrope ("net"). Immediately after the manipulation sampling started with five 36 mm inner diameter acrylic cores taken in rows along each of the structures created (cores 1 – 15), and controls randomly placed across the unaffected area (positions 16 – 20).

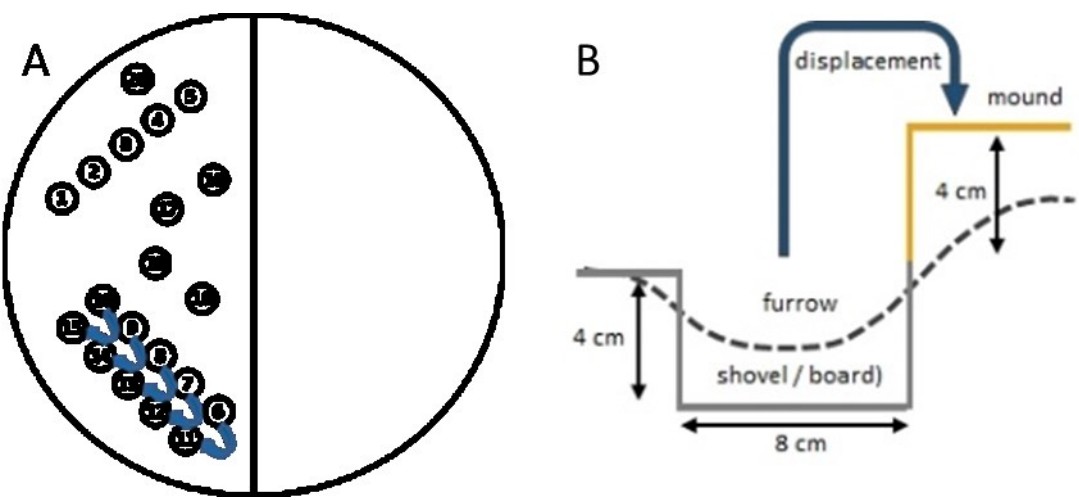

Figure 2: A: Sampling scheme on the left side of the mesocosm; a corresponding experiment on the right side including bioturbating organisms is not addressed here. Cores 1 – 5 net, 6 – 10 furrow, 11 – 15 mound, and cores 16 – 20 controls. Arrows indicate sediment removal and deposition. B: Schematic of sediment excavation. Dashed line indicates final surface topography.

### *In situ* experiment

On the 19th of June 2021 a small otter trawl typical for the area was performed at 20 m water depth at approximately 54°12'N and 11°52'E (MB 47) by RV "Solea", while RV "Limanda" and further research vessels conducted a series of associated measurements and sampling (Fig. 1). The site at 20.3 m depth is inhabited by *Arctica islandica* (46 Ind. m⁻²) in much the same way as the investigation area Fehmarn Belt (s. below). Sediments consisted of silty fine sand (m.d. 180 µm). Scuba divers sampled the net area (effect of footrope) in one trawl track and control cores were taken in the recently untrawled vicinity that same day. Five 36 mm inner diameter cores were taken randomly within 1 m² in both areas. On a second trawl track sampling by scuba diving occurred from furrow and mound areas. Here cores were inserted along the axis of the shallow furrow carved

by the otter board (Fig. 3). Material excavated by the board lay in irregular piles on the outer rim of the furrow. Cores were inserted along the highest ridge of these piles parallel to the direction of the furrow.


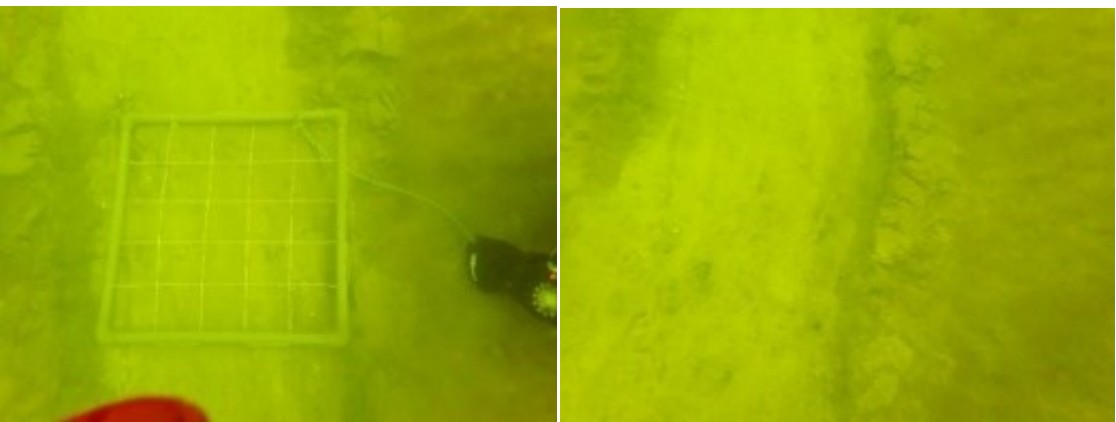

**Figure 3: UW photographs of otter board track visible as light sediment with a depth of approximately 4 cm (counting square: 50 cm width). The material displaced from the furrow is visible as mound with transversal cracks on the right hand side of the lighter furrow (right photo). Photographs curtesy of H. Pielenz.**


**Field data**

In the Fehmarn Belt area random sampling by multiple corer (MUC) was performed in 2020 and subsequent years for Chl-a depth distribution to describe the bioturbation activity. Cores (10 cm diameter) were sliced, slices homogenized and samples treated as described below.

The site is characterized by muddy fine sand (median diameter 50 µm; $C_{org}$ 5.5 % dw) (Gogina and Feldens, 2020) and is dominated by *A. islandica* (43 Ind. m$^{-2}$). Sampling of the benthic macrofauna was performed using a van Veen grab (75 kg, sieve lid) with a sampling area of 0.1 m² and sieving (0.5 mm) on 30 grab hauls. Samples were preserved with 4% formaldehyde seawater solution.

**Processing**

Processing took place within three hours after core retrieval. Sediment was carefully extruded from the tube and cut in intervals of 0.5 cm to 2 cm depth, followed by 1 cm slices to 8 cm and 2 cm slices to a final depth of 12 cm. Layers were homogenized with spatula in Petri dishes. One cm$^3$ of sediment was subsampled using a cut syringe and placed in a centrifuge tube for storage at -18 °C. For pigment extraction samples were thawed, mixed with 8 ml > 96% denatured ethanol (WALTER-CMP)

and vortexed for 5 s. They were then extracted in the dark at ~7 °C for 22 - 24 hours with one additional mixing step several hours into the extraction. Prior to the measurement samples were centrifuged at 2000 rpm for 4 minutes. Chlorophyll fluorescence was measured (Trilogy, module CHLA, Turner Designs), with additional acidification prior to the determination of phaeopigments (50 µL 1 N HCl). Chl-a concentrations ($\mu g\ cm^{-3}$) were calculated from raw fluorescence values following JGOFS Report (1994). We calibrated the fluorimeter using a spectrophotometrically determined standard curve.

Rates of Chl-a decay required for modelling particle reworking (Sun et al., 1991) were obtained by incubating (7 °C) surface sediment in the dark in re-sealable containers. During 35 days, three parallel samples were removed at different time intervals and treated as described above. From the temporal decline of Chl-a concentration a pseudo-first order constant, $k_d$, was calculated.

**Modelling data**

The distribution of chlorophyll concentrations with depth was used to infer rates of transport, separating local from non-local transport. Soetaert et al. (1996) derived a hierarchical model family consisting of six models of increasingly complex biological transport. The simplest model describes the tracer distribution without mixing by organisms and only depending on decay and sediment accumulation. In model 2 diffusive mixing is included ($D_b$ [$cm^2\ y^{-1}$]). Starting from model 3, all subsequent models
include 'non-local' transport of tracer from the sediment surface to depth with increasingly complexity (injection, J, and ingestion, r). There is no benefit in trying to differentiate between models of higher complexity (models 3, 4, 4a and 5), since fitting results seemingly differ owing to some horizontal heterogeneity in our data. Also, we lack observations that would allow to differentiate mechanisms like particle injection in physical sediment turnover and compare it to model results. Therefore, we grouped all fits of non-local models into a category "non-local" and do not interpret them further. A level of $p \leq$
0.05 was used for statistical significance in modelling (see Soetaert et al., 1996 for details).

**3 Results**

The two experiments provided similar displacement of surface sediment onto the adjacent seafloor. The effect of removal was
visible by eye as surface topography, however only with a small vertical displacement of the sediment-water interface. Peaks found in our *ex situ* experiment were made by shovelling sediment and consciously depositing it upside down onto the adjacent sediment. This created peaks at 1.25 cm depth on average. After shovelling, the sediment visibly slid to both sides of the newly formed mound, but particularly into the furrow. The relatively narrow furrow compared to the excavation depth (4 cm) might have promoted this sliding effect compared to the field experiment with a wider furrow. The Chl-a profiles imply that an

overall net deposition of 1 - 1.5 cm remained. The peak concentration (Fig. 4, upper panel) declines upwards and downwards at similar rates and approaches the background of about 8 µg cm$^{-3}$ at 3.5 cm depth. This is 2.25 cm below the original sediment water interface and corresponds well to concentrations in control and net areas below 2 cm. Concentrations in the upper one cm of the furrow are surprisingly high on average. This may be a result of sediment sliding back from the mounds.

The pattern in the *in situ* experiment is the same as *ex situ* despite very different levels of concentrations. In the field experiment, the impact by the ground net remains undetectable as net profiles resemble controls closely. Surface mean concentrations of 4 µg cm$^{-3}$ decline quickly with depth. Furrow surface concentrations are somewhat lower than controls and indicate a net decapping effect of about 1 cm in the *in situ* experiment. The peak at 1.25 cm depth in the mound resembles that in the *ex situ* experiment. With 6 µg cm$^{-3}$ its average concentration is higher than $z = 0$ concentrations in the controls and may indicate some sorting mechanism active. Morys et al. (2021) described surface sediment removal by dredging with very similar patterns in the furrow zone.

Overall, the generating mechanism and the effect of this type of otter board on the particle tracer profile appear clear. The displaced sediment generated a peak in the vertical profile of particulate chlorophyll in both experiments. The profile shape indicates that the peaks result from the fact that the sediment material was flipped over. Maximum surface concentrations of two interfaces meet at depth of about 1.25 cm (Fig. 4). This generates a pronounced signal of sediment movement. A much higher concentration close to the sediment surface in the *ex situ* experiment than at the *in situ* site resulted from the sediments rich in microphytobenthos sampled in preparation of our experiment from a shallow local site.

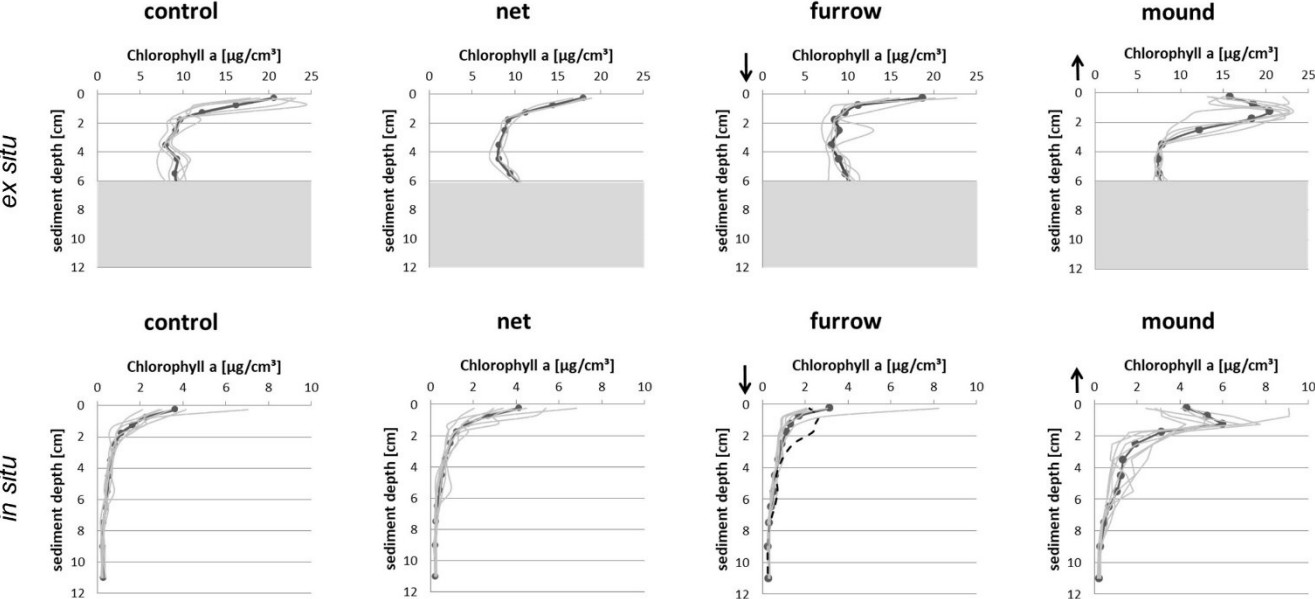

**Figure 4: Averaged (bold) and individual Chl-a tracer profiles (grey) for both *ex situ* (upper row) and *in situ* experiments (lower row). Dashed line in lower furrow panel represents one core sampled with an individual of *Arctica islandica* found at 2 cm depth. Arrows indicate that the topography of the sediment water interface was shifted relative to control or net areas by approximately 1 - 1.5 cm: down in the furrow and up in the mound area. Shaded areas in upper panels indicate pigment levels not relevant to the present discussion.**

Field data in Fehmarn Belt showed peaks of different maximum concentration and depth in 7 out of 11 profiles randomly sampled. Figure 5 depicts three examples and one profile without peak. Most peaks were positioned within the top 2 cm of the sediment with the exception of one single peak at 3 – 5 cm shown in figure 5. Such peaks are commonly interpreted as signs of non-local transport resulting from faunal activity.

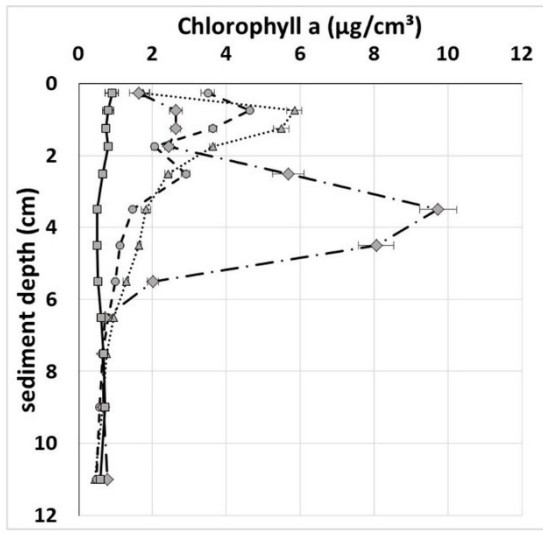


**Figure 5: Examples of peaks detected in Fehmarn Belt during random sampling by multiple corer.**

We compiled the qualitative information on non-local versus local reworking retrieved from mixing.exe by giving the number of best fits in each category (Table 1). The overview includes three field sites with muddy sediment or silty fine sands. Peaks

are a frequent feature in sediments from Mecklenburg Bay and Fehmarn Belt when sampled randomly from ships. Without explicitly targeting specific features produced by demersal fishery gear (net, mounds or furrows) 32 to 64% of samples depicted peaks. In the two experiments where trawl features were specifically sampled, peaks were most abundant in mound areas (70 – 80%). Samples from furrows displayed comparatively many profiles without any mixing, however they also showed local and non-local mixing at roughly equal numbers. Net areas and control areas were similar in that they were dominated by local

mixing in the trawl experiment. The *ex situ* control and net values in table 1 should be considered with caution because of inconsistencies in tracer distribution below 6 cm.

**Table 1: Compilation of results from targeted sampling in the two experiments, highlighting local versus non-local transport as interpreted using the software mixing.exe (Soetaert et al., 1996). Each column shows the numbers of depth-profiles modelled and**
**column 'non-local' in brackets the percentage of non-local modelling results. Our own results randomly obtained in the Fehmarn Belt area in 2020 and data reported by Morys et al. (2016) from Mecklenburg Bay are provided for comparison.**

| | local (Db) | non-local (J; r) | no mixing |
|---|---|---|---|
| **In situ Experiment** | | | |
| control (n=7) | 6 | 1 (14) | 0 |
| net (n=7) | 6 | 1 (14) | 0 |
| furrow (n=7) | 3 | 2 (28) | 2 |
| mound (n=7) | 2 | 5 (71) | 0 |
| | | | |
| **Ex situ Experiment** | | | |
| control (n=5) | 2 | 3 (60) | 0 |
| net (n=3) | 0 | 3 (100) | 0 |
| furrow (n=5) | 1 | 1 (20) | 3 |
| mound (n=5) | 1 | 4 (80) | 0 |
| | | | |
| **Field data** | | | |
| Fehmarn Belt 2020, unpublished own data (7 of 11) | 3 | 7 (64) | 1 |
| Mecklenburg Bay, Morys et al. (2016) (7 of 22) | 10 | 7 (32) | 5 |

## 4 Discussion

Two experimental approaches to sediment reworking by otter board trawling showed largely identical results. Both simulated *ex situ* sediment reworking using a shovel as surrogate and targeted sampling of *in situ* otter board marks revealed the same pattern of sub-surface peaks. This leads us to conclude, at least qualitatively, that the mechanism acting is the same. It indicates, too, that the otter board reverses a sediment slab more than it pushes sediment aside. Deposition of part of the excavated surface sediment onto the adjacent sediment-water interface generates concentration peaks of the particle tracer Chl-a. These peaks resemble those of biogenic origin in non-local bioturbation.

This similarity poses a problem in areas like Fehmarn Belt where Chl-a peaks in the vertical sediment profile may stem from bioturbating infauna or fishery gear. The uncertain origin of the observed peaks may affect the assessment of bioturbation intensity, since peaks may only indicate sediment perturbation in general and not necessarily that owed to macrofauna at the seafloor. Similarly, addressing quantitative aspects of otter board disruption of the seafloor may be difficult.

There are more or less pronounced additional peaks visible in some individual *in situ* profiles. Based on the decay of Chl-a tracer with time we are certain that any previous trawling event that may have generated the peak would be older than 3- 4 months. It would therefore not interfere with our interpretation of the relocation of fresh surface particle tracer. However, we found an *A. islandica* individual of 2 cm length at the peak depth in the sediment core from which we obtained the profile highlighted in the *in situ* furrow (Fig. 4). In view of the core diameter (36 mm) we attribute this peak directly to the activity of this animal. Patchiness in concentrations in horizontal and vertical direction is obvious in both field and *ex situ* data. This is a reminder that there may always be peaks left without known origin, and that averaging of replicate profiles is advisable.

We regularly encountered Chl-a peaks in Fehmarn Belt where MUC-sampling was random. Still 7 of 11 randomly cored locations displayed peaks at depths comparable to our experimentally generated peaks (Tab. 1; Fig. 4, 5). These peaks cannot be assigned to either trawling or bioturbation with any certainty.

Below we will pursue the questions 1) if this problem is unique to our area of investigation in Fehmarn Belt, 2) which implications it may have for our understanding of the ecosystem and 3) if there are ways to resolve it?

**Uniqueness to our area of investigation**

It is not clear if the problem of indistinguishable peaks exists in other trawling areas of the oceans, too. In the areas investigated here *A. islandica* is the benthic organism dominant in biomass (Zettler et al., 2001) and is thought responsible for reworking of sediments (Morys et al., 2016, 2017). The area is also among the most heavily trawled in the North Sea and Baltic Sea region (Amoroso et al., 2018). We are not aware of a similar comparison of trawling effects and bioturbation effects in the literature.

In general, both transport phenomena, biological non-local reworking and physical impact of the otter boards, will move sediment particles in a similar way, at least on a relatively coarse scale of centimeters as employed here. Decapping by trawling is described by Morys et al. (2021) and Bradshaw et al. (2021) who reported the removal of the top sediment layer in an artificial furrow in the Baltic off southern Sweden. While their report is based on direct observation and measured vertical distributions of sediment parameters, several indirect reports from other seas show that this decapping effect seems to be a feature observed in other trawling areas (Oberle et al., 2016b; Tiano et al., 2019; Depestele et al., 2019; Morys et al. 2021). However, the observation of a peak generated during the process of decapping, as in the present paper, has not been reported by any of the previous authors.

Oberle et al. (2016b) attribute mixing of sediments by trawl boards on the Iberian shelf as deep as 35 cm into the sediment. They deciphered trawling effects on sediments off the Iberian west coast by combining lithological information, traces from recent oil pollution and highly resolved spatial information on bottom trawling. These authors could not be certain to exactly sample the sediment site where an effect of a board would be visible. The authors suggested five scenarios of impact based on different lines of evidence. Scenario 2 is the one mirroring our turnover of surface sediment. The former sediment surface is buried and tracer appears at 18 cm rather than at 6 cm, the usual mixing depth by animals and currents in the area. At the Iberian site the large difference in depth makes a differentiation between shallow biogenic peaks and deep peaks generated by fishing boards possible.

Contrary to the situation found by Oberle et al. (2016b), the peaks of Chl-a are undistinguishable when two co-occurring mechanisms of transport generate them at similar depths. Morys et al. (2016, 2017) used a pairwise comparison of the fauna present and the occurrence of a peak within the same sediment core to convincingly argue for the animals causing that peak. The authors found positive correlations in 40-70% of the 22 investigated profiles including locations in close proximity (Mecklenburg Bay) to both our field sites. This is compelling evidence for biogenic origin of peaks. While there is little direct

proof of a cause-effect relation of biological peak generation in the literature, there is no doubt that some benthos does generate peaks by non-local transport mechanism (e.g. Blair et al., 1996). We suspect that the origin of peaks in a given area can only be inferred if bioturbating organisms, gear used and intensity of trawling are known.

**Implications**

The most notable consequence of indistinguishable peaks is likely that bioturbation studies may be suspected to overestimate the biogenic reworking effect. We will argue below (Ways to resolve) that this is not the case in Fehmarn Belt, however, this issue might need observation in other areas where trawling and bioturbation are prominent. If some peaks were indicating trawling and not non-local reworking, our concept of how the seafloor is reworked might need reassessment.

Effects within the sediment may be visible when heavy gear is used. In the Fehmarn Belt area fishing gear is comparatively light. We did not detect changes in Chl-a distribution in the net area as the simulated and real impact profiles in net areas were indistinguishable from respective controls in our study. Nets are reported to mix surface sediments and reduce Chl-a content in the top centimeter (Oberle et al., 2016b; Tiano et al., 2019). Depestele et al., (2019) working at a south-western Frisian Front site in the North Sea found that trawling may affect particle size distribution down to 2-4 cm depth. They also suggested that this trawling caused injection of finer particles into the sediment at about 4 cm depth, while winnowing the top surface sediment due to a combined mechanism of sediment removal (decapping) and mixing. While we cannot exclude similar winnowing or injection in our data, their role is likely small judging from the similarity to control depth profiles of Chl-a. Biological mixing of particles, reworking (Kristensen et al., 2012), and bioresuspension (Graf and Rosenberg, 1997) may generate the same profile in control sediment as seen in the net area. In our data there is no visible decapping effect in the net area, possibly because of the smaller size gear employed. Trawling gear referred to by Depestele et al. (2019) and used in the North Sea is in general much larger than the gear used in our study and in Fehmarn Belt. We thus conclude that the net impact is not detectable in our Chl-a depth distributions and mixing appears to be quasi random and similar to controls.

While otter boards may generate peaks and tracer profiles that resemble those of bioturbating fauna, the mechanisms of particle transport differ considerably and imply different consequences. Removing and overthrowing sediment as by scouring otter boards, can burry animals, a process which does not usually happen during bioturbation. Particularly smaller surface-dwelling benthos (cumacea, ophiurids) and meiofauna may not be able to escape from a slap of sediment deposited above them. Larger burrowers like *A. islandica* have the ability to escape, depending on the height of deposits above the original sediment surface (Bromley 1996, Powilleit et al., 2009). Some fauna may be excavated from the sediment or damaged during board passage and become available for predators and scavengers. Reports on the shift of functional groups towards smaller size and possibly opportunistic fauna caused by bottom trawling support this (Sparks-McConkey and Watling, 2001; Hiddink et al., 2019). As demonstrated by Mestdagh et al. (2018), particle reworking, possibly as a consequence of an escape reaction, and bioirrigation may mitigate effects of trawling.

Reversal of the top sediment will also affect sediment biogeochemistry, since it changes chemical gradients close to the sediment-water interface completely. In the troughs decapping locally exposes anoxic sediments to oxygenated overlying water. While oxygen consumption was not measured, it conceivably may increase when the board exposes high concentrations of reduced dissolved substances in the furrow or when reactive organic matter such as chlorophyll is buried in the mound. It may decrease, however, with less reactive sediment exposed in the furrow or with the reversed sediment slap forming the mound surface (Tiano et al., 2019, 2022). Van de Velde et al. (2018) simulated and measured complete homogenisation of a 15 cm surface layer due to mixing by trawling nets as well as dumping of a homogeneous 15 cm layer on top of an existing sediment. They found pronounced enhanced mineralisation dominated by anaerobic pathways and effects on manganese cycling as a consequence of both scenarios. Based on samples from the experiment we report here, Röser et al. (2022) suggest that the coupled Fe-Mn-P cycle reacts very sensitively, as expressed by altered porewater gradients, indicating Mn enrichment in the mound area and Mn loss in the furrow. A disruption of the steady-state biogeochemical distribution is apparent in both furrow and mound areas, although our setting differs considerably from more massive gear trawling impacts discussed for the North Sea (van de Velde et al., (2018; Tiano et al,. 2019) both in sediment height (~5 cm versus 15 cm) and mechanism (decapping/turnover versus mixing).

The present investigation cannot further explore organic carbon fate after such trawling impacts (Epstein et al., 2022) since we did not generate corresponding data. Transient redox reactions in the sediment initiated by trawling (Bradshaw et al., 2021; Morys et al., 2021; Tiano et al., 2019), however, may differ considerably from redox oscillating know to occur during bioturbation (Forster et al., 1996, Aller et al., 2014, Gilbert et al., 2016). Redox oscillations as they occur along burrows of infauna are considered a driver for differences in chemical speciation, element cycling and priming processes associated with bacterial carbon cycling. Their occurrence is intrinsically linked to a spatial diffusion geometry associated with burrows and biogenic structures (Aller 1994). Since the physics and geometry of an otter board impact on the sea floor is vastly different, we anticipate that its effects on diageneses in fact differ substantially from bioturbation effects.

In conclusion, we argue that we need to know if peaks indicate trawling or bioturbation, because their effects on biota and the way they affect geochemical processes differ substantially.

**Ways to resolve?**

In our Fehmarn Belt data set, we cannot assign a single peak with any certainty to biological or physical reworking. Can knowledge of the environment help to resolve this issue, since differences in peak generation likely indicate different effects in the ecosystem? We may obtain some certainty about the origin of peaks by employing generalized information on faunal abundance and behavior as well as trawling intensity as outlined below.

Fehmarn Belt is amongst the areas with very high fishing intensity globally (Amoroso et al. 2018). The International Council for the Exploration of the Sea (ICES) assesses trawling effort as $SAR_{subsurface}$, swept area ratio below 2 cm sediment depth (v.

Dorrien, pers. communication/2023 project report; ICES database: https://doi.org/10.17895/ ices.data.20310255.v3; Eigaard et al., 2016) with 0.1 per quarter or 10% of an area within three months. This number relates strictly to mobile bottom-contacting gear and includes specific information regarding gear and vessel size, but lacks high spatial resolution. In order to

relate this information to our data, we averaged 4 areas (size: 5.6 x 3.2 km area, i.e. 17.9 km²) from April - June 2020 overlapping with the research area in Fehmarn Belt in which we sampled and detected peaks in June 2020. Although it is difficult to relate this information to the very nature of physical scouring effects and their spatial distribution on the seafloor, it is the best information available. With $SAR_{subsurface}$ we can imagine that otter boards affect below surface sediment in 10 % of the area within this quarter (April - June 2020). We can equally imagine this as a probability: otter boards potentially

generate peaks in 0.1 m² on every m² within three months or 0.4 $m^{-2}$ $yr^{-1}$ if homogeneously spaced.

On the other hand, *A. islandica* are likely to move daily. Their abundance in Fehmarn Belt amounts to around 40 individuals $m^{-2}$. The animals usually stay close to the sediment surface because of their short siphon. They move the shells when gaping and closing in response to threat by demersal predators and food supply for filter-feeding. Ballesta-Artero et al. (2017) demonstrated that gaping activity in these bivalves changes seasonally with food supply, but shows a minimum of 1-2 gaping

events per month during lowest activity in winter. The authors could not determine the frequency of gaping during high activity phases, but it is certainly higher. We assume that shell movement during gaping is a mechanism allowing surface material including Chl-a pigments to slide along the shell into deeper sediment layers (anywhere to 4 cm depth, the size of Arctica-individuals). Forty individuals could produce at least 40 transport events per month or 480 $m^{-2}$ $yr^{-1}$. The numbers calculated suggest 3 orders of magnitude higher frequencies for biogenic non-local transport (480 versus 0.4 $m^{-2}$ $yr^{-1}$).

The spatial aspects of the particle transport events discussed above are particularly difficult to assess, since patchy occurrence of *A. islandica* and clustering of trawl tracks (Schönke et al., 2022) are frequent. Mound width and thus the area showing peaks generated by otter boards, likely depends on sediment type, steepness of the mounds and is additionally altered by the "bumpy" and discontinuous character of sediment deposition along the furrow (Morys et al., 2021). Despite this, with similar assumptions as above, and with more uncertainty we estimate that *A. islandica* rework sediment on 0.94 m² $yr^{-1}$ (5 cm diameter

circle around one animal, i.e. 19.6 cm² reworked multiplied by 480 $m^{-2}$ = 0.94 m²). This is a larger area than the area disturbed by otter boards (0.1 m²).

Thus, we consider it conservatively safe to assume that the majority of peaks detected in Fehmarn Belt stem from *A. islandica* active on a daily scale. Bioturbation by *A. islandica* in Fehmarn Belt should thus be the more frequent particle reworking process when compared to otter board sediment reworking. Therefore, we may continue to interpret chlorophyll peaks as

bioturbation traces in this area. Researchers in other areas of the oceans impacted by bottom trawling may perform similar estimates of the likelihood of trawling and bioturbation traces as shown here, if information on trawling intensity and abundance of major bioturbating fauna is available.

While we feel confident to say that we generally look at bioturbation when we find peaks, this does not imply that the effects of bottom trawling in our area are not important or may be visible in other data. The present result does not withstand impacts

which we cannot measure using the particle tracer Chl-a, such as on fauna mortality, sediment resuspension or remobilization

of reduced sediment components. Furthermore, we cannot elucidate with the present data the biological and biogeochemical effects associated. Particularly the spatial magnitude of both bioturbation and trawling need better quantification for such a comparison. Future exclusion of fishery in the area will provide a test field in which the persistence of peaks may be tested and their origin confirmed.


### Data availability

Data are presently not available from a data repository, because it seems unnecessary for the review process or for the readers to access depth profiles of chlorophyll. Figures 4 and 5 include the data necessary to validate the conclusions made in this
manuscript. Data are available from the corresponding author.  If necessary, however, we will supply the data in a repository with doi in due time.

### Author contributions

All authors provided data and contributed to writing this manuscript. CR, JL and LP sampled and measured in the experiments, MP and SF in the field data. SF and MP provided the concept and most of the figures, SF and MP wrote the manuscript.

### Competing interests

The contact author has declared that none of the authors has any competing interests

### Acknowledgements

We thank the crew of RV Limanda for their field work assistance as well as Christian v. Dorrien for the valuable discussion on quantification of trawling effort. This research was supported by the German Federal Ministry of Education and Research
(BMBF9 within the DAM pilot mission "MGF-Ostsee" (03F0848B).

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
