# Peer review of "Bottom fishery impact generates tracer peaks easily confused with bioturbation traces in marine sediments"

_Biogeosciences, 2023_

## Referee Comment (RC2)

**Review of Forster et al. 'Bottom fishery impact generates tracer peaks easily confused with bioturbation traces'**

**General comments**

With their study Forster et al. aim to qualitatively assess whether bottom trawling generates peaks in the sediment Chl.a profile that are similar to the characteristic subsurface concentration peaks that are commonly used as indicator of biological bioturbation. To answer their question, the authors conducted two experiments (one ex situ, one in situ) and in situ environmental sampling in the western Baltic Sea. With the results from the ex situ experiment mimicking trawling and the in situ experimental trawls the authors could show that peaks in the sediment Chl.a concentration profile are caused by re-deposition of Chl.a-rich surface sediment through the otter boards ('flipping sediment over'). The shape of these experimental Chl.a profile corresponds to the shape that is commonly associated with biological bioturbation and similar to the profiles detected in the sediment cores from random in situ sampling in an area that has both, bioturbating bivalves and intense bottom trawling. This highlights the question of physical or biological originality of those Chl.a peaks and the authors discuss implications and potential resolution of it. Because of the similar shape of Chl.a profiles from biological bioturbation and physical trawling in their study area, the authors put their findings into environmental context and present estimates of Chl.-a transport events from bioturbation and trawling. This discussion point is very interesting and I would have liked to read some more about their thoughts and arguments on the biogeochemical and ecological context of their findings.

The research question that the authors pose is realitively specific and 'technical' but it raises thoughts on the ecological significance of using tracers like sediment Chl.a concentration as tracer to quantify bioturbation of benthic macrofauna, especially in areas where trawling occurs. The conclusions that the authors draw are consistent with the evidence and arguments they present and they adress the main question. However, I found that the main research question and the conclusions are not very clearly formulated in the main text and not easy to identify for the reader. The combination of ex situ and in situ experiments with environmental sampling build a strong realistic picture in the results though I could not fully grasp how the modelled results (Table 1) fitted into the story. Here I would suggest to provide some more description/explanation so that readers unfamilar with Soetaert et al's models and Oberle et al's scenarios can follow. I would also be interested if there exist publications of Chl.a profiles where the peak can clearly be attributed fauna bioturbation, or have shown those in experiments and whether those could be shown/re-drawn here for comparison if data available (e.g. from Morys et al. 2017)? Overall, the manuscript is written well, but a number typos exist and I personally found the sentence structure hard to read and sentences and paragraphs lacked connection and reading flow which it is hard to identify the meaning/relevancy as a reader .

**Detailed notes**

**Title**: maybe add 'in marine sediments' to end of title, so readers know which environment this is in/about?

**Abstract**

Line 10: why indroducing local and non-local transport in the abstract when not referring to it any further in the abstract? Instead I suggest to shortly introduce what is meant by tracer peaks as this is something unfamiliar readers might not know about.

Line18: this sounds contradictory to me, as you just stated above that the peaks are not distinguishable. Please clarify what you mean here. I am assuming that when you put your findings into the environmental context of the study area then it becomes apparent that bioturbation is the dominating process generating the tracer peaks, at least in your study area.

**Introduction**

Overall, I find that the Introduction is lacking flow (paragraphs not well linked) and a clear lead towards the research question and research aim of this study. I think the section would benefit from restructuring the paragraphs and identifying what is important to introduce to readers to understand the topic and what the knowledge gap/research question is. Some aspects currently provided in the Introduction might not be relevant, or at least the relevance is not clear, while some interesting aspects around the tracer peaks and trawling effects come short (especially at line 62).

Line 21-26: I suggest to add that bottom trawling affects ecosystem functions (Epstein et al. 2021)

Line 33-34: What kind of scenarios are presented by Oberle et al. (2016b)?

I cannot see how the second paragraph about difficulties in detecting effects of fishing gear is linked to the research topic of this study. It might help to add a last sentence to that paragraph to show the connection to your research topic about tracer peaks.

Line 39: remove the '…'

Line 44: concentration profiles of what and where?

Paragraph 4: Great that tracer distribution and peaks as sign of bioturbation are introduced and explained for less familiar readers. However, I could not quite follow from line 47-51 and how this is relevant. Also the end of the paragraph does not provide a concluding sentence about how this connects to the study focus.

Line 56: '.' after (Aller, 2014) and move sentence of 'The western Baltic to next paragraph' because this is a new topic unrelated to bioturbation and tracer peaks.

Line 59: chl-a → Chl-a (as you have introduced the abbreviation). This is the case several times in the manuscript, please correct and make consistent.

Line 61: Peaks of what?

Line 62: *'However, intense trawling raises the question of an alternative origin of the peaks.'* → How so and where is the knowledge gap? Here the Introduction becomes interesting in my opinion.

Paragraph 6: how is this detailed description of A. islandica relevant to your research question?

Line 71: 1999 benthic study → reference missing

Lines72: AFDM abbreviation without explanation (I know what it means, but I am not sure all readers know?)

**Materials and methods**

Line 81-82: Combine sentences or switch them around, e.g. *We investigated the changes of particle distribution brought about by the mechanic impact of otter trawling. For this, we performed experiments ex situ in a mesocosm mimicking trawl marks to understand the underlying mechanism and in situ by setting trawl marks to provide proof of principle of mechanisms.* (Although this might be repetitive of the last paragraph of the introduction, or could be integrated there).

Line 88: make title 'Ex situ experiment'

Line 93: change to *'from the field site Schnaterman (SM, figure 1)…'* or similar, remove the *'close by?'*

Line 96: is the Corg content relevant? If not, remove, if yes try and provide a value, maybe from previous publications?

Line 106: were controls placed before or after sediment manipulation? I could imagine that the sediment manipulation with the shovel and rake resuspended fine material that may have affected the surrounding surface area from where controls were taken?

Figure 2: A and B not shown in the figure panels, only in the caption

Line 114: make title 'In situ experiment'

Line 119: Can you say anything about how sure you are that the untrawled area that you sampled hasn't been trawled by other fisher boats previously? And if yes, how long before you ran the in situ experiment?

Line 120: *'Five 36 mm inner diameter cores were taken randomly in an area of 1 m² each.'* → of the untrawled area or of the net area of the trawl or both?

Line 134: each 10cm diameter slice was homogenised, right? Slices were not mixed together, I assume.

Line 144: What type of ethanol (quality, purity, manufacturer)?

Line156: Were all models from Soetaert et al. used in your study?

Line 160: Why the grouping and no further interpretation? What is the aim for running this model in your study?

Line 161: remove the '&'

**Results**

Line 166-174: What about the profiles of control, net and furrow from fig 4? They are not mentioned in the text at all but shown in the figure.

Figure 4: Why show shaded area when not relevant for the key results of your study? This raises questions as to what happened there exactly which distracts from your key result in my opinion.

Line 193-201: Are these results (and Table 1) derived from modelling the data? To identify what type transport was present? So you did not use the model to quantify any transport rates? I think this part might need clarification.

Line 195: put 'net, mounds or furrows' into brackets, remove the '-'

Table 1: Do I understand correctly that data from Fehmarn Belt are from your study but data from Mecklenburg Bay are from Morys et al.? This is not very clear in caption and table. Also, please clarify that the numbers in the tables depict numbers of profiles.

Figures 4 and 5: Can you please provide figures with a bit higher resolution, so that they are easier to read and features come out more clearly?

**Discussion**

Overall, paragraphs are really short and not well connected lacking reading flow. Some paragraphs appear puzzled together with random statements that don't seem connected or where the link to your study results is not apparent.

Subheadings: It would be great if the subheadings already present the key message of the discussion topics, makes it easier for the reader to orientate and grasp the story.

I like the structure of the discussion in Uniqueness, Consequences, Ways to resolve, but the text in these sections does not really relate to the questions that you want to answer (Line 246-247).

Some final concluding thoughts at the end of the discussion would be great.

Line 211-247: I think a lot of this text belongs into the result sections or is repeated from result section and implications of the results are only partly touched upon. I suggest to shorten this text focusing on communcating your key results and conclusion from it and then continue with the following subsections (Uniqueness, Consequences, Ways to resolve) as these are your actual discussion.

Line 215: you mean biogenic?

Line 238: 'cannot rule out…' → what implications does this have for your results? Please discuss.

Line250-271 (Uniqueness): This sections has some interesting discussion points but I don't see that the questions of whether *'this problem is unique to our area of investigation in Fehmarn Belt'* is answered.

Line 255: you mean coarse? (found this typo more often, please check and correct)

Line 267: 'Boards and rolls keeping the net gear open.' → I don't understand how this statement is relevant here? This is a good example of a seemingly random sentence, not connected to the content/topic of a paragraph, which happens a lot throughout the manuscript. Please work on the text structure/reading flow.

Line 268: Are you talking about the furrow generated by the trawl here when referring to Morys and Bradshaw? And how is this linked to your results?

Line 270-271: This has been said several times now.

Line 274-294 (Consequences): The authors touch on consequences of bottom trawling in this section, but I thought the question to be answered was what consequences/implications it has that bioturbation-induced peaks and trawl-induced peaks are not distinguishable? I think the latter would be a more suitable discussion topic linked to the research question and key finding of this study.

Line 275-280: This information from Oberle's study needs to be given in the introduction where you refer to it for the first time and on which your study is based on or linked to (if I understood correctly). Then you can refer back to it here in the discussion.

Line 296: Ways to resolve → I read this section in the understanding that this referes to ways to resolve the issue that bioturbation effects and trawl effects can't be distinguished in tracer profile, making their interpretation harder. I like this section and would suggest to elaborate a bit more on why it is important to resolve it and what aspects should be considered in which research contexts. For example, when quantifying and extrapolating bioturbation transport rates based on Chl.a profiles, your results imply that it should be checked whether there is frequent trawling in that area where the extrapolations are done and whether trawling might cause a bias in the bioturbation quantification. I feel, especially in this section, to add a focus on ecological and biogeochemical context (What does this mean?) might work well and shape the manuscript to fit better to the aims and scope of the journal.

Line 297-305: I do not see how this is related to resolving the issue.

Line 307: This is the interesting question for this discussion section, but it needs to be made clear for the reader what the '*issue*' is.

Line 313: 0.1 of what per quarter? Frequency or area?

Line 319-320: Where did you get the probability assumption from or why do you assume this?

Line 333: So the numbers 480 vs 0.4 $m^{-2}$ $yr^{-1}$ reflect transport event per area and time? And what is your conclusion here, how can the issue be resolved?

Literature cited:

Epstein, G., J. J. Middelburg, J. P. Hawkins, C. R. Norris, and C. M. Roberts. 2022. The impact of mobile demersal fishing on carbon storage in seabed sediments. Glob Chang Biol **28**: 2875–2894. doi:10.1111/gcb.16105

---

## Author Comment (AC1)

BG-2023-145_Forster_RC1 response

Dear Dr Tiano, Referee 1.

Thank you for your stimulating comments. Generally, you will find almost all of the detailed suggestions adopted (see appended pdf). We have met your request for a more intensive discussion of the consequences that a shallow introduction of chlorophyll rich sediment, which is potentially rich in organic matter, may have for biogeochemical processes and distribution/fluxes of related substances.

We have also altered the objectives, since we do realize that the original wording may have led to a misconception. In this context we provide more background.

On the other hand, however, we cannot reduce the information given for Arctica, although you suggested to do so. This is important justification to the calculations leading to the final conclusion: peaks are more likely from fauna than from boards.

Thank you for taking the time to review.

answers to your comments:
In **bold** there is (part of) the original referee's comment,
Our response starts as indicated with >> … including reference to "new lines", which refer to "simple markup" view in MS Word.

(In the context of your comments on line 75, we added some text for clarity in lines 38)

**-50: good to mention that Chl-a tracers degrade over time.**
>> Done in new L. 52

**-55: … it would be good to state or at least give an example of explicit biogeochemical consequences of sediment mixing/movement on biogeochemistry ….**
>> now included in new L. 59: "with potential effects on carbon burial and  inorganic nutrient release, including potential feed-back of these rate changes on bioturbating macrofauna."

**-60: Better to state something about trawling being quite abundant rather than "massive" ….**
>> wording has been changed (new L. 66)

**-75: With the way the objectives are worded, I am a little concerned that the study may be perceived as having some bias. …**
>> Evidence that peaks may be generated by otter boards are indeed not frequent and some are already given in the original manuscript, such as Oberle et al. (2016b). In order to pursue your valuable comment, we added text in new L. 38 in order to introduce more the idea of complex changes in matter, biota and processes in surface sediments. In addition, we altered the objectives in new L. 82 to: "i) mimic the genesis of altered chl-a distributions experimentally, ii) compare these with peaks found in the field and potentially originating from commercial trawling and finally iii) to discuss the likelihood of confusion of these peaks with those generated by bioturbation."

**-90: "… a natural, horizontally homogenous vertical chl-a distribution"…**
>> … has been changed to: "… reconstruct a horizontally homogeneous, vertically declining chl-a distribution."

**-100: I am still wondering if this method of shoveling is truly representative of the effect from trawl doors. For me, I would imagine that trawl doors kind of 'plow' through the sediment continuously rather than shoveling discrete areas of sediment which would essentially turn the sediment upside down in the resulting mound. Are you able to add or rephrase some text to give more confidence in this part of the methodology to show that it is indeed representative of in-situ trawling effects?**
>> 'Plowing' is the image we had in mind, when we saw images from board tracks which look much like that in figure 2 in Morys et al. 2021. Slabs or clods of sediment are aligned along the furrow drawn by the dredge or board. Modern plowing on land essentially turn surface soil over in a line of slabs too, however we were not sure that this is what happens in water saturated sediments under water and whether this is the appropriate mental model. In the *ex situ* experiment we tried to keep a defined depth down to which we would impact the sediment. But we failed achieving this in the confined space available by applying some plow-like tool. Therefore, we decided to use the shoveling, which was better defined in geometry, but is less similar to the process that we imagine associated with an otter board.
We rephrased that passage to make clear that this is a surrogate and that we did not have a perfect model impact in the *ex situ* experiment. In the aftermath our results show that whatever the difference was in the two mechanisms, the results are surprisingly similar.

New text (L. 108): "We decided to excavate sediment keeping a defined geometry of depth and width when removing sediment, rather than some plough-like tool which we found difficult to implement. In fact we are not sure about the exact mechanism and geometry of sediment removal and its deposition on the adjacent sediment, therefore the method used in the mesocosm is a surrogate and not necessarily the same as the physical process that might be active *in situ*. We excavated sediment with a flat rectangular shovel, scoop by scoop to about 4 cm depth …"

We also refer to this again in the discussion (new L. 229/230), since we now feel that the difference in mechanisms and the results obtained all the same are noteworthy.

**-160: Is there a good reason why the models were not interpreted further and why model 5 was not investigated (was it just not relevant to the study)? If so, please state that for the reader.**
>> (not including model 5 was not intended; it should have read 'models 3 to 5' instead of 'models 3, 4 and 4a')
New L. 172 reads: There is no benefit in trying to differentiate between models of higher complexity (models 3, 4, 4a and 5), since fitting results seemingly differ owing to some horizontal heterogeneity in our data. Also, we lack observations that would allow to differentiate mechanisms like particle injection in physical sediment turnover and compare it to model results. Therefore, we grouped the fits of non-local models into a category "non-local" and do not interpret them further.

**-Results: I would personally prefer that certain sentences which describe results from the experiment be in past tense and that more general statements be in present tense. Really not a big deal but consider changing certain words to past tense in the results.**
>> We changed this accordingly.

**-Fig 4. It would make things much easier for the reader (especially lazy readers like myself) if the ex-situ (top row) and in-situ panels (bottom row) were explicitly stated in the figure itself rather than just**

**in the caption.**
>> the information was introduced into the figure itself

**-193: I thought the table was more 'quantitative' model output rather than qualitative (example: muddy vs sandy).** >> New L. 207: "We compiled the quantitative information on non-local and local reworking retrieved from mixing.exe by giving the number of best fits in each category (Table 1)."

**-217: You show that the results of trying to quantify vertical tracer concentrations create uncertainty for assessing bioturbation, but do they still provide insight on sediment mixing in general which may be caused from either faunal or human induced perturbations? The model results would then be more indicative of general disturbance rather than from only bioturbation.**
>> This is exactly the problem; however, the present wording does not seem to capture this sufficiently well. The distribution AND the model results may only indicate general disturbance, but cannot tell us if this is generated by humans or animal.

New L. 233: "The uncertain origin of the observed peaks may affect the assessment of bioturbation intensity, since peaks may only indicate sediment perturbation in general."

**-225: Depending on the type of sediment, you can see when previously resuspended sediment has settled on the seabed surface. Were you able to observe this visually in the furrows? If so, you can make the argument slightly stronger about "… sediment sliding back from the mounds". Also, could it not have been from re-settled sediment which was previously resuspended? That could leave smaller particles (typically higher in OM and chl-a) to resettle at a slower rate eventually settling on the seabed surface (with larger particles underneath). This was observed in an experimental study from Tiano et al., (2021) in fine sandy sediments but not in muddy sediment samples (though it may not be representative at all for what happens in this study).**
>> We have no observation of settling fluff or sediment in the furrow. We also took care when shoveling to not resuspend or resuspend as little as possible. In situ we did not observe any exceptional Chl concentrations that might indicate resuspension/settling.
We do not intend to discuss this, since it is irrelevant to our observation and therefore do not change that sentence, which is now line 245.

**-255: You can probably remove the last sentence of the page (you already state that it is irrelevant).**
>> You are right. (remove from what is now new L. 272)

**-265: In addition to nets/parts of the gear mixing sediment, Depestele et al., 2016 and 2018 suggest that this mixing occurs in tandem with the removal of surface sediment. This section would benefit with the inclusion of these studies in the discussion and their findings (erosion + mixing rather than these effects in isolation). Furthermore, De Borger et al., 2021 use this erosion + mixing effect as input for a biogeochemical model. Perhaps the findings from the current study suggest a different or at least an additional effect for modelers to take into account when predicting biogeochemical trawling impacts.**
>> New L. 279: we have added to the discussion here. The new passage reads:
"The pattern of particle tracers generated by ground trawling, however, may depend on gear and sediment type. Nets are reported to mix surface sediments and reduce chl a content in the top centimeter (Tiano et al. 2019; Oberle et al. 2016b). Depestele et al. (2019) working at a southwestern Frisian Front site in the North Sea found that Tickler-chain trawl affected particle size distribution down to 2-4 cm depth. They also suggested that this trawling caused injection of finer particles into the

sediment at about 4 cm depth, while winnowing the top surface sediment, due to a combined mechanism of sediment removal (decapping) and mixing. The trawling gear with mechanical fish chasing mechanisms used in their study is much larger than the gear used in ours. In our data there is no visible decapping effect in the net area, likely because of the smaller size gear employed. Mixing only may be the correct approach in biogeochemical modelling in contrast to de Borger et al. (2021) who used a combination of erosion and mixing effect to infer consequences for carbon mineralisation and N-cycling. In our study the simulated and real net impact profiles were indistinguishable from respective controls. This implies that sediment mixing appears to be quasi random and similar in both cases. We thus conclude that these net impacts are not detectable in chl-a depth distributions."

**-285: Mestdagh et al., (2018) found lower sediment oxygen consumption after deposition of low organic sediment as well as lower macrofaunal densities. They also found higher bioturbation and bioirrigation when low amounts of sediment were deposited and the opposite when high amounts of sediment were deposited. Could be relevant to the part of the discussion about depositional effects.**

**Lower oxygen consumption (also lower mineralization) would mean less carbon degradation and CO2 release and lower production of inorganic nutrients in the sediment. Furthermore, deposition of sediment with high levels of OM could have the opposite effect and could increase oxygen consumption due to the increased input of OM to the sediment (several unpublished experiments et al.; and it's also logical). Removal of OM from trawling has been attributed to decreases in O2 consumption and mineralization (Tiano et al., 2019, 2022) or increases (van de Velde et al., 2018) due to re-exposure of buried OM/priming. Lower mineralization might decrease fluxes of nutrients (less of them being generated in the sediment) but changing the diffusive boundary in the sediment water interface could increase nutrient fluxes (Warnken et al., 2003). Trawl-induced mixing + erosion can decrease denitrification (De Borger et al., 2021; Ferguson et al., 2020) but might also increase denitrification under conditions which maximize hypoxic areas within the sediment (De Borger et al., 2021). I state all this because I think there is more to be said about what kind of biogeochemical consequences are possible due to your results. Please try to expand a little bit on this section.**

>> We did expand and incorporated some of your suggestions in the new lines 306 - 312. Overall there is not much specific information for our experiment (though included), so many thoughts remain speculation. But at least the passage links more to current knowledge.
Furthermore, there is an expansion in the passage on geochemical effects starting with new L. 316:

"Reversal of the top sediment will affect sediment biogeochemistry, since it changes chemical gradients close to the sediment-water interface completely. In the troughs decapping locally exposes anoxic sediments to oxygenated overlying water. Van de Velde et al. (2018) simulated and measured complete homogenisation of a 15 cm surface layer due to mixing by trawling nets as well as dumping of a homogeneous 15 cm layer on top of an existing sediment. They found pronounced enhanced mineralisation dominated by anaerobic pathways and effects on manganese cycling as a consequence of both scenarios. Based on samples from the experiment we report here, Röser et al. (2022) suggest that the coupled Fe-Mn-P cycle reacts very sensitively, as expressed by altered porewater gradients, indicating Mn enrichment in the mound area and Mn loss in the furrow. A disruption of the steady-state biogeochemical distribution is apparent in both cases, although our setting differs considerably from van de Velde et al. (2018) both in sediment height (~5 cm versus 15 cm) and mechanism (decapping/turnover versus mixing). Oxygen consumption was not measured, but it may decrease with

less reactive sediment exposed in the furrow or with the reversed sediment slap forming the mound surface (Tiano et al., 2019, 2022). Oxygen consumption may increase, however, when the board exposes high concentrations of reduced dissolved substances in the furrow or when potentially reactive organic matter such as chlorophyll is buried. The present investigation cannot further explore organic carbon fate after such trawling impacts (Epstein et al. 2021) since we did not generate corresponding data. Transient redox reactions in the sediment initiated by trawling (Bradshaw et al. 2021; Morys et al. 2021; Tiano et al., 2019), however, may differ considerably from redox oscillating know to occur during bioturbation (Forster 1998, Aller 2014, Gilbert et al. 2016)."

**-325: I find Arctica super interesting. That said, there is too much written about these cool clams and it takes away from the message of this paper. Consider streamlining the Arctica sections.**
>> From a biological perspective these information is needed to justify the subsequent back-of –the envelop calculations that yield in the probability of a peak being more likely generated by biology than by trawling. Therefore, we do not agree and would like to keep this information.

**-332: The sentence discussing the quantified bioturbation (480 m$^{-2}$ per year non local transport) vs. trawling (0.4 m$^{-2}$ per year) might be one of the most applicable and interesting outcomes from this paper. Consider adding more to this discussion. As the message of this paper suggests that non-local mixing from trawling is similar with that of bioturbation, how does this quantified result relate to the main message of this manuscript (*does it put it in context* etc.)?**

**Moreover, this is a nice paper with some important results but I think the ending can be improved to showcase why we should find importance in these findings.**

>> We have expanded on the final thoughts to bioturbation versus otter boards (see text below). Maybe there is a typo in your comments or a misunderstanding: this paper ONLY suggests that non-local mixing from trawling is similar to bioturbation, not in the net area. (organic matter injection into below surface layers). The following passage should be more clear on this point.

New L. 371:

"The numbers calculated suggest 3 orders of magnitude higher frequencies for biogenic non-local transport (480 versus 0.4 m$^{-2}$ yr$^{-1}$).
The spatial aspects of the particle transport events discussed above are particularly difficult to assess, since patchy occurrence of *A. islandica* and clustering of trawl tracks (Schönke et al. 2022) are frequent. Mound width and thus the area showing peaks generated by otter boards, likely depends on sediment type, steepness of the mounds and is additionally altered by the "bumpy" and discontinuous character of sediment deposition along the furrow (Morys et al. 2021) leading to overestimates. Despite this, with similar assumptions as above and with more uncertainty, we estimate that *Arctica* rework sediment (5 cm diameter circle around one animal, i.e. 19.6 cm² x 480 m$^{-2}$ = 0.9 m²) on a larger area than the area disturbed by otter boards (0.1 m²).
Thus, we consider it conservatively safe to assume that the majority of peaks detected in Fehmarn Belt stem from *A. islandica* active on a daily scale. Bioturbation by *A. islandica* in Fehmarn Belt should thus be the more frequent particle reworking process when compared to otter board sediment reworking. Therefore, we may continue to interpret chlorophyll peaks as bioturbation traces in this area. The different mechanisms of sediment disturbance bear similarities such as the fresh organic matter injection

below the sediment surface demonstrated here. We cannot elucidate with the present data the biological and biogeochemical effects associated, particularly the spatial magnitude of both mechanisms need better quantification for such a comparison.

Future exclusion of fishery in the area will provide a test field in which the persistence of peaks may be tested and their origin confirmed."

**We added literature** that you suggested (De Borger et al. 2021; Depestele et al. 2016; Mestdagh et al. 2018; Tiano et al.  2022; van de Velde et al. 2018) and the following citations:

Epstein, G., Middelburg, J. J., Hawkins, J. P., Norris, C. R. and Roberts, C. M.: The impact of mobile demersal fishing on carbon storage in seabed sediments.  Glob Change Biol. 2022; 28:2875–2894.

Forster, S.: Spatial and temporal distribution of oxidation events occurring below the sediment water interface. P.S. Z.N.I: Marine Ecology, 17, 309-319, 1996.

Gilbert, F. Hulth, S., Grossi, V., Aller, R.C.: Redox oscillation and benthic nitrogen mineralization within burrowed sediments: An experimental simulation at low frequency. Journal of Experimental Marine Biology and Ecology, 482, 75-84, http://dx.doi.org/10.1016/j.jembe.2016.05.003, 2016.

---

## Author Response (AR1)

**BG-2023-145_Forster**

**Dear Associate Editor, dear Jack.**
**This point-to point reply is based on responses to RC1 and RC2 as well as additional comments (AC)**

I general we removed many distracting lines of discussion making the text there more concise. On the other hand, we explained some issues better (e.g. the model used, …), expanded the results presentation in the text and added discussion requested by the reviewers (geochemical effects, implication of possible confusion, …). Overall the manuscript increased in length by roughly 60 lines or 15%.

Best regards, Stefan Forster

Dear Dr Tiano, **Referee 1**.
**RC1** … We have met your request for a more intensive discussion of the consequences that a shallow introduction of chlorophyll rich sediment, which is potentially rich in organic matter, may have for biogeochemical processes and distribution/fluxes of related substances.
We have also altered the objectives, since we do realize that the original wording may have led to a misconception. In this context we provide more background.
On the other hand, however, we cannot reduce the information given for Arctica, although you suggested to do so. This is important justification to the calculations leading to the final conclusion: peaks are more likely from fauna than from boards.
…
answers to your comments:
In **bold** there is (part of) the original referee's comment,
Our response starts as indicated with >> … including reference to "new lines", which refer to "simple markup" view in MS Word.

**response to RC2**
Dear **Referee 2.**
Thank you for a thorough, critical and very constructive review. Your accurate analysis of wording and structure is particularly helpful (and impressive). It improves the message in this manuscript a lot and is much appreciated.
You understood what we did, the reasoning behind it and how we discussed our results correctly. We understand your criticism and suggestions to improve the text, and therefore include most of your suggestions clarifying the text for the reader.

Our response below is indicated by ">" and follows the original comments by the reviewer, RC2 (kept in **bold**).

**RC2 General comments**
**With their study Forster et al. aim to qualitatively assess whether bottom trawling generates peaks in the sediment chl-a profile that are similar to the characteristic subsurface concentration peaks that are commonly used as indicator of biological bioturbation. To answer their question, the authors conducted two experiments (one ex situ, one in situ) and in situ environmental sampling in the western Baltic Sea. With the results from the ex situ experiment mimicking trawling and the in situ experimental trawls the authors could show that peaks in the sediment chl-a concentration profile are caused by re-deposition of chl-a rich surface sediment through the otter boards ('flipping sediment over'). The shape of these**

experimental chl-a profile corresponds to the shape that is commonly associated with biological bioturbation and similar to the profiles detected in the sediment cores from random in situ sampling in an area that has both, bioturbating bivalves and intense bottom trawling. his highlights the question of physical or biological originality of those Chl-a peaks and the authors discuss implications and potential resolution of it. Because of the similar shape of chl-a profiles from biological bioturbation and physical trawling in their study area, the authors put their findings into environmental context and present estimates of chl-a transport events from bioturbation and trawling. This discussion point is very interesting and I would have liked to read some more about their thoughts and arguments on the biogeochemical and ecological context of their findings.

> It is positive to see that the core of what we did and intend to present in this manuscript is understandable. You will find below that we have expanded the discussion on context and consequences of peaks and combined it with the issue that arises when the peaks' genesis with respect to bioturbation or trawling is unclear.

**RC2 The research question that the authors pose is relatively specific and 'technical' but it raises thoughts on the ecological significance of using tracers like sediment Chl-a concentration as tracer to quantify bioturbation of benthic macrofauna, especially in areas where trawling occurs. The conclusions that the authors draw are consistent with the evidence and arguments they present and they address the main question. However, I found that the main research question and the conclusions are not very clearly formulated in the main text and not easy to identify for the reader. The combination of ex situ and in situ experiments with environmental sampling build a strong realistic picture in the results though I could not fully grasp how the modelled results (Table 1) fitted into the story. Here I would suggest to provide some more description/explanation so that readers unfamilar with Soetaert et al's models and Oberle et al's scenarios can follow. I would also be interested if there exist publications of Chl-a profiles where the peak can clearly be attributed fauna bioturbation, or have shown those in experiments and whether those could be shown/re-drawn here for comparison if data available (e.g. from Morys et al. 2017)? Overall, the manuscript is written well, but a number typos exist and I personally found the sentence structure hard to read and sentences and paragraphs lacked connection and reading flow which it is hard to identify the meaning/relevancy as a reader.**

> following the first reviewer (RC_1) we altered some parts of the discussion already and, for instance, integrated Epstein et al. (2021) and other additional literature.
As we outline below, and following your suggestion we now restructure the discussion while keeping its general structure with three sub-headings.

**RC2 Title: maybe add 'in marine sediments' to end of title, so readers know which environment this is in/about?**

> We include this suggestion. New title: Bottom fishery impact generates tracer peaks easily confused with bioturbation traces in marine sediments.

**ABSTRACT**
AC We omitted the word "(presumably)".

**RC2Abstract**

**RC2 Line 10: why introducing local and non-local transport in the abstract when not referring to it any further in the abstract? Instead I suggest to shortly introduce what is meant by tracer peaks as this is something unfamiliar readers might not know about.**

> We omitted the second sentence here including the truly unnecessary differentiation between local and non-local. Instead, we inserted a new one: Bioturbation (particle reworking) includes downward transport of particles into the sediment as a major process and is sometimes detected as sub-surface maxima (peaks) of specific particulate substances (tracers).

**RC2 Line 18: this sounds contradictory to me, as you just stated above that the peaks are not distinguishable. Please clarify what you mean here. I am assuming that when you put your findings into the environmental context of the study area then it becomes apparent that bioturbation is the dominating process generating the tracer peaks, at least in your study area.**

> Correct.  To clarify this, we changed the last sentence in the abstract to: However, based on natural fauna abundance, behavioural information and fishery intensity data, we identify macrofauna and not otter boards as the dominant cause for peaks at the sites investigated here.

AC we altered a sentence that ended without proper end as "… whereby." (Line 14)

**INTRODUCTION**

**RC1 -50: good to mention that Chl-a tracers degrade over time.**
>> Done in new L. 52

**RC1 -55: … it would be good to state or at least give an example of explicit biogeochemical consequences of sediment mixing/movement on biogeochemistry ….**
>> now included in new L. 59: "with potential effects on carbon burial and  inorganic nutrient release, including potential feed-back of these rate changes on bioturbating macrofauna."

**RC1 -60: Better to state something about trawling being quite abundant rather than "massive" ….**
>> wording has been changed (new L. 66)

**RC1 -75: With the way the objectives are worded, I am a little concerned that the study may be perceived as having some bias. …**
>> Evidence that peaks may be generated by otter boards are indeed not frequent and some are already given in the original manuscript, such as Oberle et al. (2016b). In order to pursue your valuable comment, we added text in new L. 38 in order to introduce more the idea of complex changes in matter, biota and processes in surface sediments. In addition, we altered the objectives in new L. 82 to: "i) mimic the genesis of altered chl-a distributions experimentally, ii) compare these with peaks found in the field and potentially originating from commercial trawling and finally iii) to discuss the likelihood of confusion of these peaks with those generated by bioturbation."

**RC2 Introduction**
**Overall, I find that the Introduction is lacking flow (paragraphs not well linked) and a clear lead towards the research question and research aim of this study. I think the section would benefit from restructuring the paragraphs and identifying what is important to introduce to readers to understand the topic and**

**what the knowledge gap/research question is. Some aspects currently provided in the Introduction might not be relevant, or at least the relevance is not clear, while some interesting aspects around the tracer peaks and trawling effects come short (especially at line 62).**

**RC2 Line 21-26: I suggest to add that bottom trawling affects ecosystem functions (Epstein et al. 2021)**
> New sentence: Trawling affects ecosystem functions such as carbon storage (Epstein et al 2021) and sediment integrity (de Juan *et al.* 2015), and additionally interacts with other pressures on the benthic ecosystem such as contaminant deposits or hypoxia (Oberle et al., 2016b; Bunke et al., 2019; van Denderen et al., 2021).

**RC2 Line 33-34: What kind of scenarios are presented by Oberle et al. (2016b)? I cannot see how the second paragraph about difficulties in detecting effects of fishing gear is linked to the research topic of this study. It might help to add a last sentence to that paragraph to show the connection to your research topic about tracer peaks.**
> This paragraph was split in two. The first one addresses that sediments may also be disrupted by other processes than trawling: "Investigations aiming to detect and quantify the effects of fishing gear at the seafloor face the difficulty that patterns may also stem from disturbances, natural or anthropogenic, other than trawling (Bunke et al., 2019). …… "
> A new paragraph starting "Localization of the impact on the sea floor and … " focusses on the problems associated with localizing precisely the trawling marks in order to then sample precisely there, whatever is needed to assess the effects trawling has. "Measures such as swept area surface (SAR) of fishing intensity remain inaccurate in that they average bottom trawls over long periods (per year or per quarter) and relate the impact to comparatively large areas (several km²)".
A final sentence has already been added in response to RC_1. "However, the damage to the surface sediment by otter boards may be local. Studies investigating the change of vertical distribution of sediment constituents demonstrate that an effect of trawling can be the removal of surface sediment and considerable alterations of matter concentrations and processes in this sediment surface layer (Mestdagh et al 2018; Morys et al., 2021; van de Velde et al. 2018). In an experimental dredge trawl Morys et al. (2021) found that sediment excavated to 2.5-3 cm depth piled up irregularly on the sides of the track."
We are now more explicate as to why we introduce these problems.

**RC2 Line 39: remove the '…'.**
> Done

**RC2 Line 44: concentration profiles of what and where?**
> The sentence has been changed to: " … and creates peaks in vertical concentration profiles of particles in the sediment (e.g. chlorophyll used as a particle tracer).

**RC2 Paragraph 4: Great that tracer distribution and peaks as sign of bioturbation are introduced and explained for less familiar readers. However, I could not quite follow from line 47-51 and how this is relevant. Also the end of the paragraph does not provide a concluding sentence about how this connects to the study focus.**
> We have reduced the detail of information and separated the remaining information into separate sentences. This makes the issue of decay in tracer more understandable: "Peaks are observed regardless of the persistence of the tracers used, in stable tracers or tracers that decay or degrade with time. Natural decay of the tracer chlorophyll allows to "look back in time" for 100 - 150 days when its peak concentration has declined to 25 % of its original value. Decay thus determines whether a peak will remain visible or if the event merges into the overall mixing which is usually dealt with as diffusion analogue." In the preceding sentence the terms stable and decaying replaced experimental and natural, respectively, therefore the line of thought should be clear now.

A concluding sentence states that "Chlorophyll as a particle tracer can therefore show relatively recent events of particle mixing only", which is the introductory information needed here (natural, decaying tracer with about 1/3 of a year time frame looking back.

**RC2 Line 56: '.' after (Aller, 2014) ok and move sentence of 'The western Baltic to next paragraph' because this is a new topic unrelated to bioturbation and tracer peaks.**
> Yes, the sentence is moved. Subsequent sentence changed to fit.

**RC2 Line 59: chl-a ⬚ Chl-a (as you have introduced the abbreviation). This is the case several times in the manuscript, please correct and make consistent.**
> Throughout the manuscript there will only be "Chl-a"

**RC2 Line 61: Peaks of what?**
> Now changed to "Chlorophyll peaks detected …

**RC2 Line 62: 'However, intense trawling raises the question of an alternative origin of the peaks.' -> How so and where is the knowledge gap? Here the Introduction becomes interesting in my opinion.**
> We agree that this point may need introduction. In conjunction with the next comment to paragraph 6 we moved the sentence towards the end of the introduction and after paragraph 6 (on *Arctica*), expanding the thought as a separate paragraph "under water video evidence.
(new L.77) "However, intense trawling raises the question of an alternative origin of the peaks. Previous studies (Oberle et al., 2016b), UW-video evidence, and a possible analogy to terrestrial soil turnover during ploughing triggered the idea of surface sediment subduction by otter boards. To our knowledge there is no evidence of this transport process in the literature."
This also helps to explain how we developed the idea that sediments might be moved in this particular way in the first place.

**RC2 Paragraph 6: how is this detailed description of A. islandica relevant to your research question?**
(new L.68) > We shortened this paragraph by 4 lines removing information about depth, age and preferred sediment, thereby restricting the information to the *Arctia* to the minimum necessary in the context of bioturbation.

**RC2 Line 71: 1999 benthic study -> reference missing**
> done

**RC2 Lines72: AFDM abbreviation without explanation (I know what it means, but I am not sure all readers know?)**
> done

**MATERIALS & METHODS**

**RC1** (In the context of your comments on line 75, we added some text for clarity in lines 38)

**RC1 -90: "… a natural, horizontally homogenous vertical chl-a distribution"…**
>> … has been changed to: "… reconstruct a horizontally homogeneous, vertically declining chl-a distribution."

**RC1 -100: I am still wondering if this method of shoveling is truly representative of the effect from trawl doors. For me, I would imagine that trawl doors kind of 'plow' through the sediment continuously rather than shoveling discrete areas of sediment which would essentially turn the sediment upside down in the resulting mound. Are you able to add or rephrase some text to give more confidence in this part of the methodology to show that it is indeed representative of in-situ trawling effects?**

>> 'Plowing' is the image we had in mind, when we saw images from board tracks which look much like that in figure 2 in Morys et al. 2021. Slabs or clods of sediment are aligned along the furrow drawn by the dredge or board. Modern plowing on land essentially turn surface soil over in a line of slabs too, however we were not sure that this is what happens in water saturated sediments under water and whether this is the appropriate mental model. In the *ex situ* experiment we tried to keep a defined depth down to which we would impact the sediment. But we failed achieving this in the confined space available by applying some plow-like tool. Therefore, we decided to use the shoveling, which was better defined in geometry, but is less similar to the process that we imagine associated with an otter board.

We rephrased that passage to make clear that this is a surrogate and that we did not have a perfect model impact in the *ex situ* experiment. In the aftermath our results show that whatever the difference was in the two mechanisms, the results are surprisingly similar.

New text (L. 106): "We decided to excavate sediment keeping a defined geometry of depth and width when removing sediment, rather than some plough-like tool which we found difficult to implement. In fact we are not sure about the exact mechanism and geometry of sediment removal and its deposition on the adjacent sediment, therefore the method used in the mesocosm is a surrogate and not necessarily the same as the physical process that might be active *in situ*. We excavated sediment with a flat rectangular shovel, scoop by scoop to about 4 cm depth …"

We also refer to this again in the discussion (new L. 242), since we now feel that the difference in mechanisms and the results obtained all the same are noteworthy.

**RC1 -160: Is there a good reason why the models were not interpreted further and why model 5 was not investigated (was it just not relevant to the study)? If so, please state that for the reader.**

>> (not including model 5 was not intended; it should have read 'models 3 to 5' instead of 'models 3, 4 and 4a')

New L. 171 reads: "There is no benefit in trying to differentiate between models of higher complexity (models 3, 4, 4a and 5), since fitting results seemingly differ owing to some horizontal heterogeneity in our data. Also, we lack observations that would allow to differentiate mechanisms like particle injection in physical sediment turnover and compare it to model results. Therefore, we grouped the fits of non-local models into a category "non-local" and do not interpret them further."

**RC2 Materials and methods**
**RC2 Line 81-82: Combine sentences or switch them around, e.g. We investigated the changes of particle distribution brought about by the mechanic impact of otter trawling. For this, we performed experiments ex situ in a mesocosm mimicking trawl marks to understand the underlying mechanism and in situ by setting trawl marks to provide proof of principle of mechanisms. (Although this might be repetitive of the last paragraph of the introduction, or could be integrated there).**

**RC2 Line 88: make title 'Ex situ experiment'**
> done, and likewise for "In situ experiment" below

**RC2 Line 93: change to 'from the field site Schnaterman (SM, figure 1)...' or similar, remove the 'close by?'**
> done

**RC2 Line 96: is the Corg content relevant? If not, remove, if yes try and provide a value, maybe from previous publications?**
> Removed, because irrelevant. This information on Corg is also removed from "In situ experiment"

**RC2 Line 106: were controls placed before or after sediment manipulation? I could imagine that the sediment manipulation with the shovel and rake resuspended fine material that may have affected the surrounding surface area from where controls were taken?**
(new L. 117) > Controls were taken afterwards and this is now pointed out in the text. Since we did not see any resuspension effect, such as sedimentation of Chl-a on those controls, we will also point this out in the result section.

**RC2 Figure 2: A and B not shown in the figure panels, only in the caption**
> A and B are shown in the figures now, too.

**RC2 Line 114: make title 'In situ experiment'**
> done

**RC2 Line 119: Can you say anything about how sure you are that the untrawled area that you sampled hasn't been trawled by other fisher boats previously? And if yes, how long before you ran the in situ experiment?**
> Based on information by the federal fishery agency (Thünen Institute for Baltic Sea Research, Rostock) there was little activity in the study area. Our preceding geophysical acoustic survey showed few tracks compared to e.g. Fehmarn Belt.
Admittedly, surveys of fisher boats in the Baltic are far from perfect. And geoacoustic information might also miss tracks. However, we know that tracks that are on the order of a few months old always show up clearly in acoustic images.
For Chl-a as a tracer to work, our track needed to be fresh and we sampled this fresh material that was displaced. Older peaks from older tracks should have been documented in peaks with reduced Chl-concentrations due to the decay of chlorophyll with age. We cannot see any significant peaks in our individual profiles that would indicate an event producing a peak within the last 1-2 months (50% surface concentration) or 3-4 months old (25% surface concentration).
Thus we are certain that any possible previous trawling does not interfere with our interpretation of the relocation of fresh surface particle tracer, as shown in our Chl-a peaks.
We only added the word "recently".

**RC2 Line 120: 'Five 36 mm inner diameter cores were taken randomly in an area of 1 m² each.' -> of the untrawled area or of the net area of the trawl or both?**
> Each was meant to indicate that both areas were sampled; changed to make the information clear.

**RC2 Line 134: each 10cm diameter slice was homogenized, right? Slices were not mixed together, I assume.**
> Correct, now changed to homogenized

**RC2 Line 144: What type of ethanol (quality, purity, manufacturer)?**
> The information is now provided: > 96% denatured ethanol (WALTER-CMP)

**RC2 Line 156: Were all models from Soetaert et al. used in your study?**

> No they were not. Text now includes more information, also because RC1 asked for it: "Starting from model 3, all subsequent models include 'non-local' transport of tracer from the sediment surface to depth with increasingly complexity (injection, J, and ingestion, r). There is no benefit in trying to differentiate between models of higher complexity (models 3, 4, 4a and 5), since fitting results seemingly differ owing to some horizontal heterogeneity in our data. Also, we lack observations that would allow differentiating mechanisms like particle injection in physical sediment turnover and comparing it to model results. Therefore, we grouped the fits of non-local models into a category "non-local" and do not interpret them further".

**RC2 Line 160: Why the grouping and no further interpretation? What is the aim for running this model in your study?**

> The reason to run models is to arrive at a contrasting differentiation between non-local and local transport. This differentiation is qualitative, but arriving there in an objective procedure, by modelling, provides us with quantitative information on the transport intensity as well, which is not used in this manuscript. While this has been a bit confusing, it is now clear by our use of the term "qualitative" in the text in contexts of information that we provide.
The information now included (see answer to line 156 above) hopefully also clarifies this issue.

**RC2 Line 161: remove the '&'**
> done

**RESULTS**

**RC1 -Results: I would personally prefer that certain sentences which describe results from the experiment be in past tense and that more general statements be in present tense. Really not a big deal but consider changing certain words to past tense in the results.**
>> We changed this accordingly.

**RC1 -Fig 4. It would make things much easier for the reader (especially lazy readers like myself) if the ex-situ (top row) and in-situ panels (bottom row) were explicitly stated in the figure itself rather than just in the caption.**
>> the information was introduced into the figure itself.

**RC1 -193: I thought the table was more 'quantitative' model output rather than qualitative (example: muddy vs sandy).**
>> New L. 225: "We compiled the quantitative information on non-local and local reworking retrieved from mixing.exe by giving the number of best fits in each category (Table 1)."

**RC2 Results**
**RC2 Line 166-174: What about the profiles of control, net and furrow from fig 4? They are not mentioned in the text at all but shown in the figure.**
> This is true and we added to the description of results a little (l. 191 and

**RC2 Figure 4: Why show shaded area when not relevant for the key results of your study? This raises questions as to what happened there exactly which distracts from your key result in my opinion.**
> In order to avoid such discussion, we changed the grey field to not transparent and removed "elevated pigment levels generated by mishandling during preparation" from the figure heading. Removing all data that are not strictly relevant to the study would require truncating all eight tiles in this figure at 6 cm depth. This looks strange and raises the question if we really measured such shallow concentration profiles to any person active in the field. Therefore, we prefer to hide something without addressing it and only stating "Shaded areas in upper panels indicate pigment levels not relevant to the present discussion". This hopefully keeps further question at a minimum.
We added *ex situ* and *in situ* in the figure to make this clearer and easily visible for the reader.

**RC2 Line 193-201: Are these results (and Table 1) derived from modelling the data? To identify what type transport was present? So you did not use the model to quantify any transport rates? I think this part might need clarification.**
> Yes, they are. We had to employ the model quantifying transport rates in order to arrive at the qualitative differentiation (through the model) between local and non-local. This information is now included in the first sentence: "We compiled the qualitative information on non-local versus local reworking retrieved from mixing.exe by giving the number of best fits in each category (Table 1)."

**RC2 Line 195: put 'net, mounds or furrows' into brackets, remove the '-'**
> done

**RC2 Table 1: Do I understand correctly that data from Fehmarn Belt are from your study but data from Mecklenburg Bay are from Morys et al.? This is not very clear in caption and table. Also, please clarify that the numbers in the tables depict numbers of profiles.**
> You understood correctly, but this was truly not clear. We changed it accordingly in Table 1.

**RC2 Figures 4 and 5: Can you please provide figures with a bit higher resolution, so that they are easier to read and features come out more clearly?**

**DISCUSSION**

**RC1 -217: You show that the results of trying to quantify vertical tracer concentrations create uncertainty for assessing bioturbation, but do they still provide insight on sediment mixing in general which may be caused from either faunal or human induced perturbations? The model results would then be more indicative of general disturbance rather than from only bioturbation.**
>> This is exactly the problem; however, the present wording does not seem to capture this sufficiently well. The distribution AND the model results may only indicate general disturbance, but cannot tell us if this is generated by humans or animal.

New L. 248: "The uncertain origin of the observed peaks may affect the assessment of bioturbation intensity, since peaks may only indicate sediment perturbation in general …"

**RC1 -225: Depending on the type of sediment, you can see when previously resuspended sediment has settled on the seabed surface. Were you able to observe this visually in the furrows? If so, you can make the argument slightly stronger about "… sediment sliding back from the mounds". Also, could it**

**not have been from re-settled sediment which was previously resuspended? That could leave smaller particles (typically higher in OM and chl-a) to resettle at a slower rate eventually settling on the seabed surface (with larger particles underneath). This was observed in an experimental study from Tiano et al., (2021) in fine sandy sediments but not in muddy sediment samples (though it may not be representative at all for what happens in this study).**

>> We have no observation of settling fluff or sediment in the furrow. We also took care when shoveling to not resuspend or resuspend as little as possible. In situ we did not observe any exceptional Chl concentrations that might indicate resuspension/settling.

We do not intend to discuss this, since it is irrelevant to our observation and therefore do not change that sentence, which is now line 190 as the entire passage was removed from discussion into results on request of RC2.

**RC1 -255: You can probably remove the last sentence of the page (you already state that it is irrelevant).**

>> You are right. We removed the sentence addressing "irrelevant observations".

**RC1 -265: In addition to nets/parts of the gear mixing sediment, Depestele et al., 2016 and 2018 suggest that this mixing occurs in tandem with the removal of surface sediment. This section would benefit with the inclusion of these studies in the discussion and their findings (erosion + mixing rather than these effects in isolation). Furthermore, De Borger et al., 2021 use this erosion + mixing effect as input for a biogeochemical model. Perhaps the findings from the current study suggest a different or at least an additional effect for modelers to take into account when predicting biogeochemical trawling impacts.**

>> we have added to the discussion here.

AC  In the further process we also altered the passage, streamlined the text and shortened some passages compared to our previous RC1 response.: The passage in new line 300 now reads: "Effects within the sediment may be visible when heavy gear is used. In the Fehmarn Belt area fishing gear is comparatively light. We did not detect changes in Chl-a distribution in the net area as the simulated and real impact profiles in net areas were indistinguishable from respective controls in our study. Nets are reported to mix surface sediments and reduce Chl-a content in the top centimeter (Oberle et al., 2016b; Tiano et al., 2019). Depestele et al., (2019) working at a south-western Frisian Front site in the North Sea found that trawling may affect particle size distribution down to 2-4 cm depth. They also suggested that this trawling caused injection of finer particles into the sediment at about 4 cm depth, while winnowing the top surface sediment due to a combined mechanism of sediment removal (decapping) and mixing. While we cannot exclude similar winnowing or injection in our data, their role is likely small judging from the similarity to control depth profiles of Chl-a. Biological mixing of particles, reworking (Kristensen et al., 2012), and bioresuspension (Graf and Rosenberg, 1997) may generate the same profile in control sediment as seen in the net area. In our data there is no visible decapping effect in the net area, possibly because of the smaller size gear employed. Trawling gear referred to by Depestele et al. (2019) and used in the North Sea is in general much larger than the gear used in our study and in Fehmarn Belt. We thus conclude that the net impact is not detectable in our Chl-a depth distributions and mixing appears to be quasi random and similar to controls."

AC Upon rearranging the discussion/Implications, we removed discussion on modeling citing de Borger et al. (2021) and, similarly, on the self-priming possibility of suspended matter by van de Velde et al. (2018) as well. We now consider these modelling aspects related to net impact much too distracting from the line of discussion.

**RC1 -285: Mestdagh et al., (2018) found lower sediment oxygen consumption after deposition of low organic sediment as well as lower macrofaunal densities. They also found higher bioturbation and bioirrigation when low amounts of sediment were deposited and the opposite when high amounts of sediment were deposited. Could be relevant to the part of the discussion about depositional effects. RC1 Lower oxygen consumption (also lower mineralization) would mean less carbon degradation and CO2 release and lower production of inorganic nutrients in the sediment. Furthermore, deposition of sediment with high levels of OM could have the opposite effect and could increase oxygen consumption due to the increased input of OM to the sediment (several unpublished experiments et al.; and it's also logical). Removal of OM from trawling has been attributed to decreases in O2 consumption and mineralization (Tiano et al., 2019, 2022) or increases (van de Velde et al., 2018) due to re-exposure of buried OM/priming. Lower mineralization might decrease fluxes of nutrients (less of them being generated in the sediment) but changing the diffusive boundary in the sediment water interface could increase nutrient fluxes (Warnken et al., 2003). Trawl-induced mixing + erosion can decrease denitrification (De Borger et al., 2021; Ferguson et al., 2020) but might also increase denitrification under conditions which maximize hypoxic areas within the sediment (De Borger et al., 2021). I state all this because I think there is more to be said about what kind of biogeochemical consequences are possible due to your results. Please try to expand a little bit on this section.**

>> We did expand and incorporated some of your suggestions in what are the new lines 324-345. Overall there is not much specific information for our experiment (though included), so many thoughts remain speculation. But at least the passage links more to current knowledge.

Furthermore, there is an expansion in the passage on geochemical effects starting with new L. 324: "Reversal of the top sediment will affect sediment biogeochemistry, since it changes chemical gradients close to the sediment-water interface completely. While oxygen consumption was not measured, it conceivably may increase when the board exposes high concentrations of reduced dissolved substances in the furrow or when reactive organic matter such as chlorophyll is buried in the mound. …

This continues in that we further address oxygen consumption, manganese cycling, disruption of steady-state profiles and redox oscillations. The final sentence is: "In conclusion, we argue that we need to know if peaks indicate trawling or bioturbation, because their effects on biota and the way they affect geochemical processes differ substantially."

**RC1 -325: I find Arctica super interesting. That said, there is too much written about these cool clams and it takes away from the message of this paper. Consider streamlining the Arctica sections.**

>> From a biological perspective these information is needed to justify the subsequent back-of –the envelop calculations that yield in the probability of a peak being more likely generated by biology than by trawling. Therefore, we do not agree and would like to keep this information.

**RC1 -332: The sentence discussing the quantified bioturbation (480 m$^{-2}$ per year non local transport) vs. trawling (0.4 m$^{-2}$ per year) might be one of the most applicable and interesting outcomes from this paper. Consider adding more to this discussion. As the message of this paper suggests that non-local mixing from trawling is similar with that of bioturbation, how does this quantified result relate to the main message of this manuscript (*does it put it in context* etc.)?**

**RC1 Moreover, this is a nice paper with some important results but I think the ending can be improved to showcase why we should find importance in these findings.**

>> We have expanded on the final thoughts to bioturbation versus otter boards (see text below). Maybe there is a typo in your comments or a misunderstanding: this paper ONLY suggests that non-local

mixing from trawling is similar to bioturbation, not in the net area. (organic matter injection into below surface layers). The following passage should be more clear on this point.

New L. 371:

"The numbers calculated suggest 3 orders of magnitude higher frequencies for biogenic non-local transport (480 versus 0.4 m$^{-2}$ yr$^{-1}$).

The spatial aspects of the particle transport events discussed above are particularly difficult to assess, since patchy occurrence of *A. islandica* and clustering of trawl tracks (Schönke et al. 2022) are frequent. Mound width and thus the area showing peaks generated by otter boards, likely depends on sediment type, steepness of the mounds and is additionally altered by the "bumpy" and discontinuous character of sediment deposition along the furrow (Morys et al. 2021) leading to overestimates. Despite this, with similar assumptions as above and with more uncertainty, we estimate that *Arctica* rework sediment (5 cm diameter circle around one animal, i.e. 19.6 cm² x 480 m$^{-2}$ = 0.9 m²) on a larger area than the area disturbed by otter boards (0.1 m²).

Thus, we consider it conservatively safe to assume that the majority of peaks detected in Fehmarn Belt stem from *A. islandica* active on a daily scale. Bioturbation by *A. islandica* in Fehmarn Belt should thus be the more frequent particle reworking process when compared to otter board sediment reworking. Therefore, we may continue to interpret chlorophyll peaks as bioturbation traces in this area.

Researchers in other areas of the oceans impacted by bottom trawling may perform similar estimates of the likelihood of trawling and bioturbation traces as shown here, if information on trawling intensity and abundance of major bioturbating fauna is available.

We altered the subsequent passage reported to RC1 to: "While we feel confident to say that we generally look at bioturbation when we find peaks, this does not imply that the effects of bottom trawling in our area are not important or may be visible in other data. The present result does not withstand impacts which we cannot measure using the particle tracer Chl-a, such as on fauna mortality, sediment resuspension or remobilization of reduced sediment components. Furthermore, we cannot elucidate with the present data the biological and biogeochemical effects associated. Particularly the spatial magnitude of both bioturbation and trawling need better quantification for such a comparison. Future exclusion of fishery in the area will provide a test field in which the persistence of peaks may be tested and their origin confirmed."

**RC2 Discussion**

**RC2 Overall, paragraphs are really short and not well connected lacking reading flow. Some paragraphs appear puzzled together with random statements that don't seem connected or where the link to your study results is not apparent.**

**Subheadings: It would be great if the subheadings already present the key message of the discussion topics, makes it easier for the reader to orientate and grasp the story. I like the structure of the discussion in Uniqueness, Consequences, Ways to resolve, but the text in these sections does not really relate to the questions that you want to answer (Line 246-247).**

> In conjunction with responses to Reviewer 1 and along the lines of your suggestions we have added substantially to several aspects of the discussion, such as which implications does the undefined origin of peaks have. We remove some distracting parts and rearrange (i.e. description of results into "results") in order to streamline the reading flow. Keeping the general structure with "Uniqueness, Consequences, and Ways to resolve" has been a guideline though. Please find more details below.

**RC2 Some final concluding thoughts at the end of the discussion would be great.**

> These remarks had to be placed there, also following your last comment.

New line 382: "Thus, we consider it conservatively safe to assume that the majority of peaks detected in Fehmarn Belt stem from *A. islandica* active on a daily scale. Bioturbation by *A. islandica* in Fehmarn Belt

should thus be the more frequent particle reworking process when compared to otter board sediment reworking. Therefore, we may continue to interpret chlorophyll peaks as bioturbation traces in this area. The different mechanisms of sediment disturbance bear similarities such as the fresh organic matter injection below the sediment surface demonstrated here. We cannot elucidate with the present data the biological and biogeochemical effects associated, particularly the spatial magnitude of both mechanisms need better quantification for such a comparison. Future exclusion of fishery in the area will provide a test field in which the persistence of peaks may be tested and their origin confirmed. "

**RC2 Line 211-247: I think a lot of this text belongs into the result sections or is repeated from result section and implications of the results are only partly touched upon. I suggest to shorten this text focusing on communicating your key results and conclusion from it and then continue with the following subsections (Uniqueness, Consequences, Ways to resolve) as these are your actual discussion.**
> We removed all of the descriptive passages relating to peaks and concentrations in ex situ and in situ experiments (2 paragraphs) reducing this first part of the discussion to <60% of its original length. This content has been added to the results section in a shortened version.

**RC2 Line 215: you mean biogenic?**
> Yes.

**RC2 Line 238: 'cannot rule out…' -> what implications does this have for your results? Please discuss.**
> For the reasons given above (half-life of the Chl-a tracer, information on little previous activity, no visible peaks that could be interpreted given the background of decay time), we are certain that any possible previous trawling does not interfere with our interpretation of the relocation of fresh surface particle tracer, as shown in our Chl-a peaks. Since we removed this passage from the discussion, however, it is no longer contained in the new version of the manuscript.

**RC2 Line 250-271 (Uniqueness): This sections has some interesting discussion points but I don't see that the questions of whether 'this problem is unique to our area of investigation in Fehmarn Belt' is answered. (**new lines 265-293)
> I order to answer that question, we reduced the text to the following sequence of topics: (1) in the literature only decapping by trawling is described, (2) no publications on peaks generated directly from trawling, (3) thus we do not know of any comparisons like ours. (4) Indirect evidence for peaks from trawling comes from e.g. by Oberle et al. (2016b) when, as in their case, tracer and trawling activity were locally tightly coupled AND peaks existed at a sediment depth exceeding that of bioturbation, but reached by large trawling boards. (In a way Oberle et al. (2016b) investigated an area with conditions allowing to draw conclusions based on different lines of evidence). Following this we state, as before, (5) that if peaks are at similar depth, as in our case, their origin remains indistinguishable.
"Uniqueness", which is now called "Uniqueness to our area of investigation" now starts with "It is not clear if the problem of indistinguishable peaks exists in other fishing areas of the oceans, too." It now ends with "We suspect that the origin of peaks in a given area can only be inferred if the bioturbating organisms, the gear used and intensity of fishing are known."

**AC** (new lines 278-285): in the above mentioned context (4) we rephrased the discussion of results by Oberle et al. (2016b) substantially.

Following RC1 and to make the line of thought clearer, a larger passage was moved from "uniqueness" to the section "implications". It discusses the effect that we see in our data for the fishing net area. We cannot

see any effect in our data but random mixing, but we still discuss this in relation to what others have seen with heavier gear, however shortened, as RC 1 suggested.

**RC2 "Line 255: you mean coarse? (found this typo more often, please check and correct)**
> Yes we do. We corrected two such typos.

**RC2 Line 267: 'Boards and rolls keeping the net gear open.' -> I don't understand how this statement is relevant here? This is a good example of a seemingly random sentence, not connected to the content/topic of a paragraph, which happens a lot throughout the manuscript. Please work on the text structure/reading flow.**
> We completely removed this passage.

**RC2 Line 268: Are you talking about the furrow generated by the trawl here when referring to Morys and Bradshaw? And how is this linked to your results?**
> We did, but this information is now incorporated in the discussion of uniqueness, addressing that in association with trawling, decapping has been directly observed (e.g. by those authors) but no peak generation.

**RC2 Line 270-271: This has been said several times now.**
> Now omitted.

**RC2 Line 274-294 (Consequences): The authors touch on consequences of bottom trawling in this section, but I thought the question to be answered was what consequences/implications it has that bioturbation-induced peaks and trawl-induced peaks are not distinguishable? I think the latter would be a more suitable discussion topic linked to the research question and key finding of this study.** (new lines 295-347)
> We put a new line into this discussion, focusing first on the implications that each of the different ways that peaks may be generated have. Following this we discuss the consequences for our understanding and description of the ecosystem, if those peaks are not distinguishable. In this context we retain some aspects of the geochemical/biological consequences that each of the process have (biological and physical peak generation), but we compare and contrast them subsequently.
AC This section includes a shortened discussion on the fact that we cannot see an effect in our data for the fishing net area. (removed from "uniqueness"; s. above)
In the process we rearranged and inserted a lot of text. Track-changes here were no longer readable. Therefore, we pasted the new text into the document. The previous version including some changes simply following RC1 is still visible at the end of "Implications" track-changes.
"Implications/consequences" now starts with "The most notable consequence of indistinguishable peaks is likely that bioturbation studies may be suspected to overestimate the biogenic reworking effect." We here note that "below (Ways to resolve) we will argue that this is not the case in Fehmarn Belt, however, this issue needs observation in other areas where trawling and bioturbation are prominent…" The massage now ends with "In conclusion, we argue that we need to know if peaks indicate trawling or bioturbation, because their effects on biota and the way they affect geochemical processes differ substantially. "

**RC2 Line 275-280: This information from Oberle's study needs to be given in the introduction where you refer to it for the first time and on which your study is based on or linked to (if I understood correctly). Then you can refer back to it here in the discussion.**
> True, this is now used in "uniqueness" and introduced accordingly in the Introduction.

**RC2 Line 296: Ways to resolve -> I read this section in the understanding that this refers to ways to resolve the issue that bioturbation effects and trawl effects can't be distinguished in tracer profile, making their interpretation harder. I like this section and would suggest to elaborate a bit more on why it is important to resolve it and what aspects should be considered in which research contexts. For example, when quantifying and extrapolating bioturbation transport rates based on Chl-a profiles, your results imply that it should be checked whether there is frequent trawling in that area where the extrapolations are done and whether trawling might cause a bias in the bioturbation quantification. I feel, especially in this section, to add a focus on ecological and biogeochemical context (What does this mean?) might work well and shape the manuscript to fit better to the aims and scope of the journal.** (new lines 350-394)

> "ways to resolve" now starts with "In our Fehmarn Belt data set, we cannot assign a single peak with any certainty to biological or physical reworking. Can knowledge of the environment help to resolve this issue, since differences in peak generation likely indicate different effects in the ecosystem?"
It now ends with "Therefore, we may continue to interpret chlorophyll peaks as bioturbation traces in this area. "

**RC2 Line 297-305: I do not see how this is related to resolving the issue.**
> we omit this passage on half-life and step size.

**RC2 Line 307: This is the interesting question for this discussion section, but it needs to be made clear for the reader what the 'issue' is.**
> The "issue" is now "that we cannot assign a single peak with any certainty to biological or physical reworking.

**RC2 Line 313: 0.1 of what per quarter? Frequency or area?**
> 0.1 per quarter implies that the area swept is 0.1 m² per m² in 3 months (quarter year). This information is given 6 lines later, but we include an explanation following "0.1 per quarter" as 10% probability for any area.

**RC2 Line 319-320: Where did you get the probability assumption from or why do you assume this?**
> This is related to 0.1 which is now explained as 10% probability

**RC2 Line 333: So the numbers 480 vs 0.4 m-2 yr-1 reflect transport event per area and time? And what is your conclusion here, how can the issue be resolved?**
> There is now an altered final sentence: "The numbers calculated suggest 3 orders of magnitude higher frequencies for biogenic non-local transport (480 versus 0.4 $m^{-2} yr^{-1}$)."
And, yes, the result of this calculation (obtained for THIS AREA ONLY) shows that peaks are predominantly of biological origin and therefore consequences discussed above are mostly related to bioturbation. And for Fehmarn Belt we may continue to interpret chlorophyll peaks as bioturbation traces.

**Overall, we added literature** suggested by the reviewers:

Depestele et al. 2016
Mestdagh et al. 2018
Tiano et al. 2022

and additionally:

Epstein, G., Middelburg, J. J., Hawkins, J. P., Norris, C. R. and Roberts, C. M.: The impact of mobile demersal fishing on carbon storage in seabed sediments.  Glob Change Biol. 2022; 28:2875–2894.

Forster, S.: Spatial and temporal distribution of oxidation events occurring below the sediment water interface. P.S. Z.N.I: Marine Ecology, 17, 309-319, 1996.

Gilbert, F. Hulth, S., Grossi, V., Aller, R.C.: Redox oscillation and benthic nitrogen mineralization within burrowed sediments: An experimental simulation at low frequency. Journal of Experimental Marine Biology and Ecology, 482, 75-84, http://dx.doi.org/10.1016/j.jembe.2016.05.003, 2016.

De Juan, S., Hewitt, J., Thrush, S., Freeman, D.: Standardising the assessment of Functional Integrity in benthic ecosystems. Journal of Sea Research 98, 33-41. https://doi.org/10.1016/j.seares.2014. 06.001, 2015.

**Literature omitted**:

Schiffers et al. (2011)

---

## Author Response (AR2)

Dear Associate Editor Dr Middelburg, dear Jack

I am pleased to read this. We have taken care of all the minor aspects that you asked to change.

You will also find a doi-address included under data accessibility; we are preparing a data file with the depth profiles this publication refers to, which we will upload during the next days and that will be accessible through our university server system.

[l. 400: I suggest that you submit the Chl-a depth profiles to an international database such as Pangaea so that others can access the data.]

Thanks for enduring the long waiting process for a second reviewer.

Sincerely,

Stefan Forster